# Behavior control of membrane-less protein liquid condensates with metal ion-induced phase separation

Kibeom Hong [1], Daesun Song [1] & Yongwon Jung [1✉]

Phase separation of specific biomolecules into liquid droplet-like condensates is a key mechanism to form membrane-less organelles, which spatio-temporally organize diverse biochemical processes in cells. To investigate the working principles of these biomolecular condensates as dynamic reaction centers, precise control of diverse condensate properties is essential. Here, we design a strategy for metal ion-induced clustering of minimal protein modules to produce liquid protein condensates, the properties of which can be widely varied by simple manipulation of the protein clustering systems. The droplet forming-minimal module contains only a single receptor protein and a binding ligand peptide with a hex-ahistidine tag for divalent metal ion-mediated clustering. A wide range of protein condensate properties such as droplet forming tendency, droplet morphology, inside protein diffusivity, protein recruitment, and droplet density can be varied by adjusting the nature of receptor/ ligand pairs or used metal ions, metal/protein ratios, incubation time, binding motif variation on recruited proteins, and even spacing between receptor/ligand pairs and the hexahistidine tag. We also demonstrate metal-ion-induced protein phase separation in cells. The present phase separation strategy provides highly versatile protein condensates, which will greatly facilitate investigation of molecular and structural codes of droplet-forming proteins and the monitoring of biomolecular behaviors inside diverse protein condensates.

[1] Department of Chemistry, Korea Advanced Institute of Science and Technology, Daejeon 34141, Korea. ✉email: ywjung@kaist.ac.kr

Eukaryotic cells utilize diverse subcellular compartments to coordinate countless biochemical reactions in space and time. Recently, in addition to well-known membrane-bound organelles, the body of examples of cellular compartments without discrete enclosing membranes has been rapidly growing[1,2]. Examples include p-bodies and stress/germ granules in the cytoplasm, as well as nucleoli and Cajal bodies in the nucleus[3–7]. These so called membrane-less organelles are enriched with a distinct set of biomolecules and are also termed biomolecular condensates. Most cellular condensates exhibit remarkable liquid droplet-like behaviors. Droplet formation is reversible, and round shape droplets can fuse each other. Moreover, inside protein components are highly diffusive and can be exchanged with the surroundings. All these properties of membrane-less organelles are believed to allow spatiotemporal compartmentalization and controlled condensation of specific sets of biomolecules. Among many components of membrane-less organelles (often over hundreds of types), a small number of proteins are primarily responsible for condensate formation (termed scaffolds), and many other components are preferentially recruited to condensates (termed clients)[8–11].

A growing body of studies indicate that multivalent interactions between scaffold proteins drive liquid–liquid phase separation (LLPS) of biomolecular solutions, leading to liquid-like condensate droplet formation[11–14]. Intrinsically disordered proteins (IDPs)/regions, which are structurally undefined and often contain low-complexity sequences, are major scaffold proteins of many intracellular liquid condensates[15–21]. Studies indicate that phase separation of these condensates is driven by multiple charge–charge, dipole–dipole, and π–cation interactions among stretches of low-complexity amino acid residues in IDPs[22–25]. In addition, multivalent protein–protein interactions between structurally more defined modular binding domains and their ligand motifs also drive LLPS to generate protein liquid droplets[26–29]. The droplet formation tendency is generally enhanced when the multivalency of tandemly repeated binding modules is increased[26,30]. RNAs, which are found in various membrane-less organelles and often contain multiple protein-binding sites, also trigger various biomolecular phase separations[19,31–33]. In general, protein LLPS processes are heavily influenced by the interaction multivalency and affinity, as well as the solubility of scaffold proteins[11,12,14].

Biomolecular condensates formed with various scaffolds under diverse conditions have displayed widely varied droplet behaviors, and these behaviors are likely crucial for the condensate's specific functions as cellular compartments. Condensate engineering by editing scaffold components provides key information to understand fundamental principles governing condensate properties. Simplified model systems have been used for more precise analysis of droplet-scaffold relations, rather than using naturally occurring systems with high complexity[8,18,23,24,30,33–36]. Various IDPs and their residue mutant variants have been used to elucidate the roles of specific sets of residues as condensate-forming scaffolds[16,18,24,25,37]. For example, extensive mutational studies with fused in sarcoma family IDPs indicated that multivalent interactions between tyrosine and arginine residues govern phase separation, while serine, glutamine, and glycine residues control condensate rigidity[23,24]. On the other hand, condensate models of tandemly repeated modular proteins contributed to identifying the effects of multivalent binding valency, and affinity on both condensate assembly and regulation of condensate compositions[8,26,27,30]. Stoichiometry control of two modular scaffolds revealed that client recruitment depends on the client valency, as well as the mixed scaffold ratio[8].

However, comprehensive molecular and structural codes of scaffolds that govern diverse physicochemical behaviors of protein liquid condensates (e.g., phase separation propensity, inside diffusivity, and client enrichment) are still not well understood. More importantly, precise behavior control of protein droplet models, which is critical to investigate how diverse biochemical reactions are organized inside droplets, has not been feasible. Here, we report metal ion-induced protein phase separation to produce protein condensates with minimal scaffold modules, which allow extensive scaffold variation, and thereby precise and easy control of dynamic droplet behaviors. A single-component scaffold is composed of only a single-binding receptor domain, a ligand peptide, and a hexahistidine tag (6His). Metal ion-mediated scaffold clustering via metal-6His coordination increases scaffold valency, subsequently leading to condensate formation (Fig. 1a). Protein liquid condensates are successfully formed with four different receptor/ligand pairs. Condensate diffusivity can be varied by the nature of metal ions ($Cu^{2+}$, $Zn^{2+}$, $Co^{2+}$, and $Ni^{2+}$), incubation time, and protein/metal ion ratios. Client protein recruitment can also be controlled by varying client–scaffold interaction elements. We also observe that mutations on few scaffold residues (or even single-residue mutations) drastically alter droplet formation processes. In addition, condensate morphology, diffusivity, and scaffold density can also be varied by adding different linker motifs between the receptor/ligand pair and 6His. Lastly, we demonstrate that the present versatile protein condensates can also be formed in living cells by treating scaffold-expressing cells with $Zn^{2+}$.

## Results

**6His–metal interactions drive condensate formation of minimal protein scaffolds.** While many protein LLPS systems have been reported based on diverse IDPs, here we utilized defined interactions between modular protein receptor–peptide ligands to construct behavior controllable protein condensates. We envisioned that modular proteins will allow more precise manipulation of LLPS-driving scaffold properties, such as binding valency, affinity, and geometry, compared to structurally disordered IDPs. Among several multivalent modular domain/ligand systems that drive LLPS, we first employed the second SH3 domain from Nck (SH3) and its binding proline-rich motif (PRM). LLPS by mixing tandem repeats of SH3 ($(SH3)_n$) and PRM ($(PRM)_n$) has been well characterized[8,26]. Furthermore, binding structures of SH3/PRM are well known[38], and multiple variants with varied binding affinities are available[26,34] for potential scaffold manipulation. Although protein condensates can be formed by mixing repeated SH3 and PRM proteins (e.g., $(SH3)_5 + (PRM)_5$), we searched for more simplified SH3-PRM scaffold modules for facile scaffold variation and subsequent condensate manipulation. While systematically reducing required polymeric SH3 and PRM components for LLPS (from $(SH3)_5$ and $(PRM)_5$), we serendipitously discovered that a simple 6His-tagged SH3 and PRM fusion protein (PRM-SH3-6His) can alone phase separate into liquid droplets in the presence of $NiCl_2$ (Fig. 1a, b). A turbidity increase by protein droplet formation was observed with PRM-SH3-6His and added $Ni^{2+}$ in a metal concentration-dependent manner (Fig. 1c). Addition of ethylenediaminetetraacetic acid (EDTA), which strongly chelates various transition metal ions, including divalent nickel ion[39], to the turbid PRM-SH3-6His solution resulted in droplet deformation and decreased turbidity, again in an [EDTA]-dependent manner (Fig. 1b, d).

High-valent polymeric SH3 and PRM repeats (e.g., $(SH3)_5 + (PRM)_5$) were needed for effective droplet formation by multivalent interactions[26]. It was therefore surprising that a single SH3-PRM fusion protein formed protein condensates. Since $Ni^{2+}$ can be coordinated with multiple histidine residues simultaneously[40], we hypothesized that $Ni^{2+}$ clustered multiple (likely two)

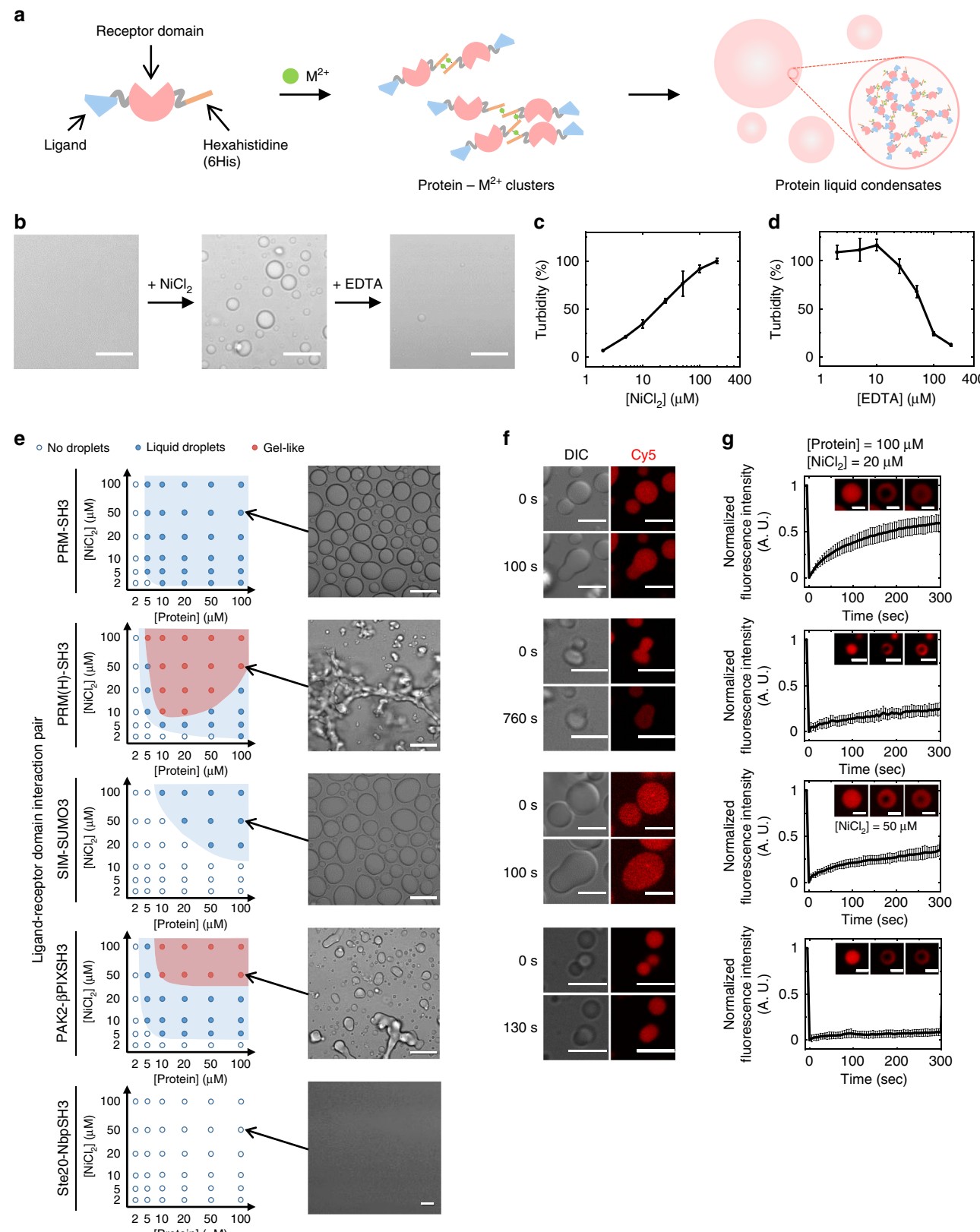

PRM-SH3-6His proteins, resulting in increased PRM-SH3 valency and subsequent condensate formation (Fig. 1a). When various other divalent metal ions were tested, only $Co^{2+}$, $Cu^{2+}$, $Ni^{2+}$, and $Zn^{2+}$, which are known to form more stable complexes with histidines in physiological pH than other metal ions, such as $Mn^{2+}$, $Mg^{2+}$, and $Ca^{2+}$ (refs. [41,42]), successfully induced phase

separation of proteins (Supplementary Fig. 1). In addition, 6His-removed PRM-SH3 proteins did not form droplets with any added metals. These data strongly support that 6His-tagged protein clustering by metal ions drives LLPS.

We next examined if metal ion-induced LLPS of 6His-tagged receptor–ligand fusion scaffolds can be applicable to other

**Fig. 1 Metal ion-induced protein phase separation. a** Schematic illustration of metal ion-induced phase separation into protein liquid condensates with a single scaffold construct (ligand–receptor–hexahistidine). **b** Optical observation of droplet formation by $NiCl_2$ addition to PRM-SH3-6His and droplet dissolution by EDTA addition. Scale bars: 10 μm. **c** Turbidity changes of a PRM-SH3-6His solution with increased $NiCl_2$ concentration. Error bars: 1 s.d. ($n = 3$). **d** Turbidity changes of a $Ni^{2+}$-induced PRM-SH3-6His condensate solution with increased EDTA concentration. Error bars: 1 s.d. ($n = 3$). **e** Phase diagrams of ligand–receptor scaffold proteins as a function of $NiCl_2$ and scaffold protein concentrations. Optical images of protein condensates at 100 μM protein and 50 μM $Ni^{2+}$ (analysis after 24 h upon metal addition) are shown in the right. Scale bars: 10 μm. **f** Optical and fluorescence images of protein droplet fusion. Scale bars: 5 μm. **g** FRAP recovery profiles and images of scaffold proteins inside condensates (analysis after 1 h upon metal addition). Scale bars: 2 μm.

protein–ligand binding pairs besides SH3-PRM (Supplementary Fig. 2). Previously, polymeric repeats of PRM(H), another ligand of SH3 with a stronger affinity than PRM, formed protein condensates upon mixing with polymeric SH3 (ref. [26]). Here, PRM(H)-SH3-6His also successfully phase separated into liquid condensates upon $Ni^{2+}$ addition (Fig. 1e). However, nonspherical and more amorphous gel-like condensates were observed at high $Ni^{2+}$ and protein concentrations (Supplementary Fig. 3), possibly due to stronger PRM(H)-SH3 interaction. Protein fluorescence intensities were rather heterogeneous for these gel-like condensates (Supplementary Fig. 4a). Human SUMO3 and the SUMO interaction motif (SIM) from PIASx, which also have shown LLPS properties[8], were next examined. SIM-SUMO-6His was also able to form protein condensates, while requiring higher $Ni^{2+}$/protein concentrations (Fig. 1e). Previously, higher numbers of SUMO/SIM repeats were needed for LLPS than SH3/PRM[8], consistent with our observation that higher metal and SIM-SUMO-6His concentrations are needed for LLPS than PRM-SH3-6His. We also tested two other (SH3 family) ligand–domain pairs with unknown phase separation ability: PAK2-βPIXSH3 (ref. [43]) and Ste20-NbpSH3 (ref. [44]). Like PRM(H)-SH3-6His, PAK2-βPIXSH3-6His formed liquid condensates and also more gel-like condensates at high concentrations (Supplementary Figs. 3 and 4a), while LLPS was not observed with Ste20-NbpSH3 at any conditions (Fig. 1e and Supplementary Fig. 3).

Dynamic droplet fusion, which is one of the representative liquid-like properties of protein liquid condensates, was observed for all formed protein condensates (Fig. 1f). Dye (cyanine 5; Cy5)-labeled proteins were also clearly enriched inside droplets (Supplementary Table 1). In addition, protein diffusivity inside droplets was measured with fluorescence recovery after photobleaching (FRAP) analyses (Fig. 1g). Proteins (100 μM) and $NiCl_2$ (20 μM) were mixed and incubated for 60 min before FRAP. PRM-SH3-6His and SIM-SUMO-6His droplets exhibited clear fluorescence recovery from bleached areas, where mobile fractions were 64% for PRM-SH3-6His and 35% for SIM-SUMO-6His (Supplementary Table 2). On the other hand, PRM(H)-SH3-6His and PAK2-βPIXSH3-6His droplets showed low mobile fractions (27% for PRM(H)-SH3-6His and 8% for PAK2-βPIXSH3-6His). Low protein mobility might be related to their tendency to form more gel-like condensates. In fact, proteins inside these gel-like condensates were nearly immobile (Supplementary Fig. 4b). We termed more spherical condensates as liquid droplets and nonspherical amorphous condensates as gel-like structures. Considering these results together, our 6His-tagged receptor–ligand system provides minimal LLPS protein modules for diverse binding pairs, which can allow facile and precise variations on condensate-forming scaffolds. In addition, the use of small metal ions to drive LLPS can grant maximal effects of scaffold variations on formed condensate behaviors.

**Metal ions for protein clustering tunes physicochemical properties of condensates.** We next investigated the effects of added metals on physicochemical properties of formed protein

condensates. The required critical metal concentration for phase separation of PRM-SH3-6His (100 μM) was lowest for $Ni^{2+}$ (2 μM), and highest for $Co^{2+}$ and $Zn^{2+}$ (20 μM; Fig. 2a). $Ni^{2+}$ provides strong binding to 6His, while $Co^{2+}$ and $Zn^{2+}$ weakly interact with His[42], possibly explaining the strong LLPS tendency by $Ni^{2+}$. On the other hand, $Cu^{2+}$ is known to have an even stronger affinity to His than $Ni^{2+}$ (ref. [42]). However, previous studies indicated that $Cu^{2+}$-His coordination is thermodynamically more stable, but kinetically more labile (fast dissociation) than $Ni^{2+}$-His coordination[45,46], which might explain the higher critical $Cu^{2+}$ concentration (10 μM). Kinetic stability of metal-induced PRM-SH3-His clusters might also be critical for phase separation. Nonetheless, all metal ions strongly condensed 6His-tagged scaffold proteins (Supplementary Fig. 5). Protein diffusivity inside condensates was also widely varied by added metal ions (100 μM PRM-SH3-6His and 100 μM metal ions). Fluorescence recoveries of bleached areas in PRM-SH3-6His droplets were fastest with $Cu^{2+}$ and slowest with $Ni^{2+}$ (Fig. 2b and Supplementary Table 3). Again, kinetically unstable $Cu^{2+}$-His interactions might contribute to this high droplet diffusivity of $Cu^{2+}$-protein condensates. In fact, when hydrogels were formed through His-divalent metal ion coordination in a previous report, $Cu^{2+}$-based gel relaxation was faster than $Ni^{2+}$-based gel relaxation[46]. Relatively immobile (rigid) $Ni^{2+}$-, $Zn^{2+}$-, and $Co^{2+}$-induced PRM-SH3-6His condensates also exhibited more resistance against condensate dissolution by EDTA than $Cu^{2+}$ condensates (Supplementary Fig. 6).

Protein clustering by metal ions before LLPS was examined by dynamic light scattering (DLS). Highly soluble green fluorescent protein (GFP) was fused to PRM-SH3-6His to provide a larger protein size for reliable DLS analysis and to prevent phase separation during measurements by increased solubility. Under the DLS analysis condition (without a crowding reagent), GFP-PRM-SH3-6His did not undergo LLPS even at 100 μM protein and 500 μM $Ni^{2+}$ (Supplementary Fig. 7). It is possible that GFP with its relatively large protein size (~25 kDa compared to PRM-SH3 ~10 kDa) might also inhibit protein interactions for LLPS. The average size of GFP-PRM-SH3-6His (50 μM) was clearly increased as the added $Ni^{2+}$ concentration was increased (from 5 μM to 200 μM; Fig. 2c). Protein size increases by clustering reached a near maximum at a 1:1 protein/metal ratio. Previous studies with model poly-His peptides suggested a stable complex formation between two 6His peptides and two $Ni^{2+}$ (refs. [40,47]). PRM-SH3-6His dimers might be the most dominant form upon $Ni^{2+}$-induced clustering. Still, multivalent interactions between PRM-SH3 will also contribute to size increases by protein chain expansion, since GFP-6His without PRM-SH3 showed smaller size increases by $Ni^{2+}$ addition (Supplementary Fig. 8a). For other metals (particularly for $Co^{2+}$ and $Zn^{2+}$), DLS size increases were significantly smaller than $Ni^{2+}$ (Supplementary Fig. 8b), consistent with the data showing that more $Co^{2+}$ and $Zn^{2+}$ were required for LLPS than $Ni^{2+}$ (Fig. 2a). The average size of PRM-SH3-6His without GFP fusion increased over 2 μm by $Ni^{2+}$ addition and presumably subsequent protein condensate formation (Supplementary Fig. 8c).

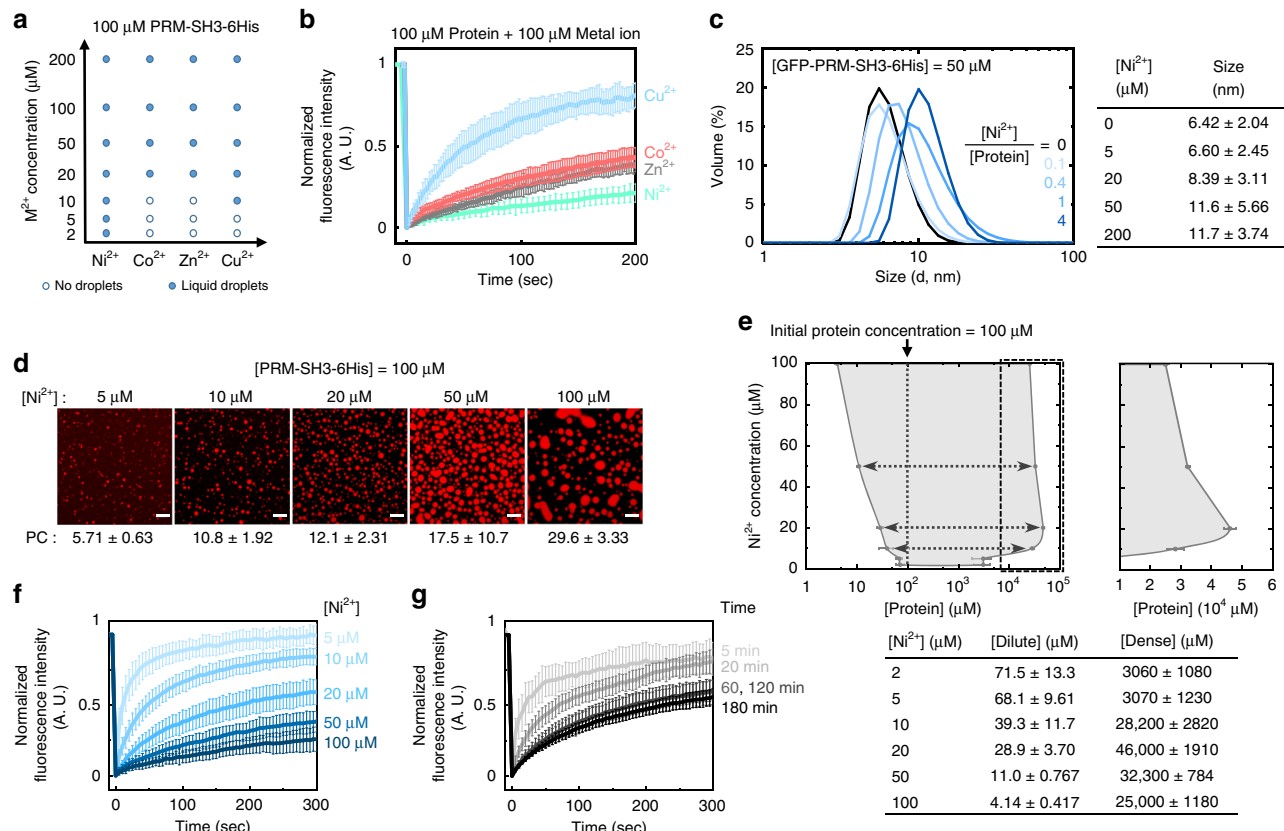

**Fig. 2 Metal ion-dependent variation of protein phase separation. a** A phase separation diagram of PRM-SH3-6His as a function of metal ion concentrations for four different metal ions. **b** FRAP recovery profiles of PRM-SH3-6His inside condensates with different metal ions. **c** DLS size distribution profiles of GFP-PRM-SH3-6His with varying ratios of $Ni^{2+}$. Average sizes for different $Ni^{2+}$ concentrations are indicated in the right table. s.d. from $n = 3$. **d** Fluorescence images of PRM-SH3-6His condensates with varying $Ni^{2+}$ concentrations (analysis after 1 h upon metal addition). Scaffold PC values are indicated below. Scale bars: 10 μm. **e** The phase diagram of PRM-SH3-6His with dense and dilute phase protein concentrations (analysis after 24 h upon metal addition). The left arm exhibits dilute phase concentrations, and the right arm for dense phase concentrations. The magnified version of the right arm (dash box in the left diagram) is drawn (right diagram). Protein concentrations were listed in a below table. Error bars: 1 s.d. ($n = 3$). **f** FRAP recovery profiles of PRM-SH3-6His inside condensates with different $Ni^{2+}$ concentrations. **g** FRAP recovery profiles of PRM-SH3-6His inside condensates with different incubation time. All FRAP analyses were conducted after 1 h upon metal addition. Error bars: 1 s.d. ($n = 3$).

We further examined liquid condensates of PRM-SH3-6His (100 μM) with varying ratios of $Ni^{2+}$ (5–100 μM). Fluorescence intensity ratios between droplet to bulk phases (partition coefficients, PCs) were higher with higher $[Ni^{2+}]$ (Fig. 2d), likely due to less proteins in bulk phases. To obtain more quantitative information on this droplet property, we determined the exact protein concentrations of droplet dense phases ($C_{dense}$) and bulk dilute phases ($C_{dilute}$) at varying $[Ni^{2+}]$ from measured fluorescence intensities. As expected from the protein PC values (Fig. 2d), $C_{dilute}$ decreased as $[Ni^{2+}]$ increased (Fig. 2e). On the other hand, $C_{dense}$ rapidly increased as $[Ni^{2+}]$ increased, but peaked (up to 46 mM) at $[Ni^{2+}]/[protein] = 0.2$, and then $C_{dense}$ slowly decreased as $[Ni^{2+}]$ increased. The ratio between clustered PRM-SH3-6His (by $Ni^{2+}$) and free PRM-SH3-6His will increase with increased $[Ni^{2+}]$. The data suggest that instead of clustering all PRM-SH3-6His with high $[Ni^{2+}]$, a certain level of free PRM-SH3-6His is needed to have a maximal condensate protein density in our multicomponent LLPS system (metal ion, clustered PRM-SH3-6His, and free PRM-SH3-6His). A recent study also reported nonlinear changes of $C_{dense}$ upon increases of component proteins in multicomponent LLPS systems[48].

Interestingly, protein diffusivity was more linearly decreased by increasing $[Ni^{2+}]$ (Fig. 2f and Supplementary Fig. 9). Droplets with 5 μM $Ni^{2+}$ and 100 μM PRM-SH3-6His showed nearly full fluorescent signal recovery, while only 20% signals were recovered

inside droplets with 100 μM $Ni^{2+}$. It is possible that condensate diffusivities are strongly governed by interaction strengths between scaffold proteins and less by the protein density in condensates. More clustered PRM-SH3-6His proteins with high $[Ni^{2+}]/[protein]$ ratios will have stronger multivalent interactions, leading to slowed protein diffusion. These data might also explain earlier condensate structure changes from liquid droplets to gel-like structures with high $[Ni^{2+}]$ for strongly interacting PRM(H)-SH3 and PAK2-SH3 proteins (Fig. 1e). To further examine this diffusivity and density changes by added metal ions, we also determined $C_{dilute}$ and $C_{dense}$ with $Zn^{2+}$ (Supplementary Fig. 10), which interacts more weakly to His and formed more mobile condensates than $Ni^{2+}$ (Fig. 2b). With $Zn^{2+}$, $C_{dense}$ was even slightly higher than those of $Ni^{2+}$ condensates, particularly at high metal concentrations, again supporting the idea that the diffusivity is more heavily influenced by scaffold interactions than densities in the present system.

After LLPS occurred by mixing PRM-SH3-6His (100 μM) and $Ni^{2+}$ (20 μM), we also examined formed condensates at different time points. Condensate PCs steadily increased over time (Supplementary Fig. 11a). In addition, condensate densities ($C_{dense}$) increased severalfold during incubation from 1 to 24 h at various $[Ni^{2+}]$ (Supplementary Fig. 11b). Interestingly, however, the condensate diffusivity was slowly decreased over 1 h (Fig. 2g) but remained constant after 1 h over 12 h at various

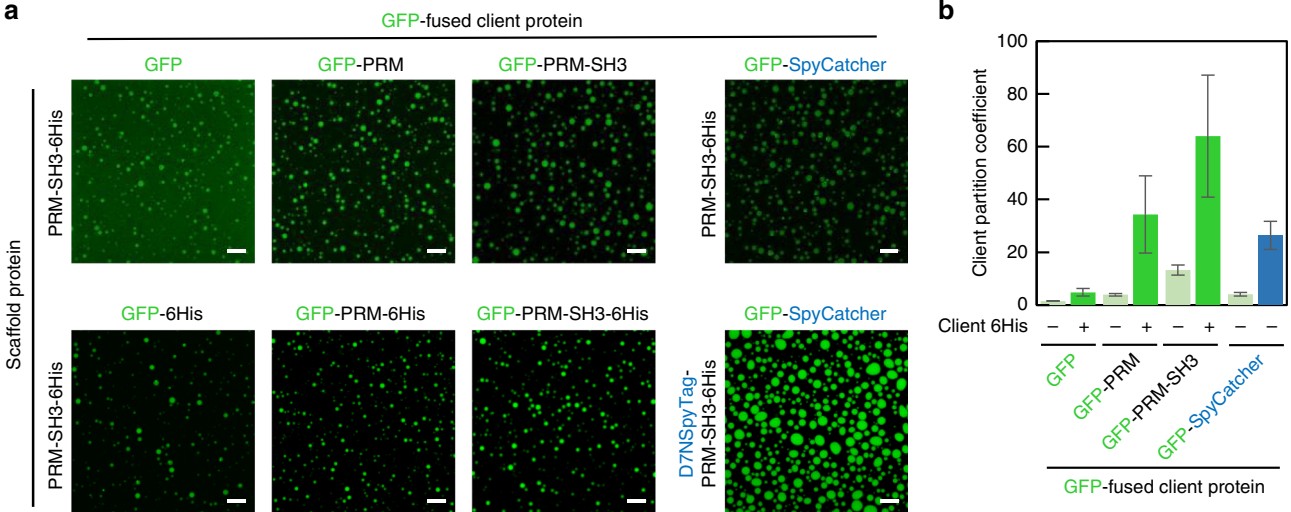

**Fig. 3 Client recruitment control into protein condensates. a** Fluorescence images of GFP-fused client recruitment into PRM-SH3-6His (or D7NSpyTag-PRM-SH3-6His) condensates. Scaffold proteins (100 μM) and client proteins (3 μM) were mixed with Ni$^{2+}$ (20 μM), and incubated for 60 min before confocal analysis. Scale bars: 10 μm. **b** Client PC values for various GFP-fused clients. PC on D7NSpyTag-PRM-SH3-6His condensates is indicated with a blue bar. Error bars: 1 s.d. ($n = 3$).

[Ni$^{2+}$] (Supplementary Fig. 11c). It is possible that interactions between metal ions and proteins, which affect protein clustering and consequent binding affinities, might reach an equilibrium within 1 h. On the other hand, condensate fusions and tight protein packing in condensates might require more time. Multiple studies reported that various condensates composed of IDPs can mature into more rigid (and/or morphologically different) droplets over time (also called aging)[12,17,18,37]. Protein condensates were also treated with EDTA after different incubation (aging) times (Supplementary Fig. 12a). Condensates were clearly disassembled (or shrunk) by EDTA, but protein droplets were still visible even after 12 h EDTA treatment, particularly for 24 h-incubated condensates. When these differently incubated condensates were diluted in phosphate-buffered saline (PBS), they were mostly resistant against PBS dilution (Supplementary Fig. 12b). Further studies will be needed to understand how more structurally defined PRM-SH3 condensates showed this time-dependent diffusivity and density changes.

**Manipulation of interacting motifs tunes client recruitment of condensates.** An ability to recruit various client biomolecules into condensates is a key characteristic for protein condensates to act as membrane-less organelles. In general, clients could be recruited into condensates by adding condensate-forming scaffold components to clients or applying other orthogonal binding pairs to scaffolds and clients[8,35,37]. For example, (SH3)$_4$ + (PRM)$_4$ droplets recruited SH3 or PRM containing clients, and adding multimeric SH3 or PRM to client further enhanced recruitment[8]. Providing quantitative recruitment degrees of various client constructs is important for the proper use of newly developed LLPS systems. Therefore, we also examined whether client recruitment could be controlled with the present metal-induced condensates of minimal PRM-SH3-6His scaffold modules. In addition to SH3 and PRM, 6His can also be used for client recruitment. Client GFP was fused with PRM, SH3, and/or 6His, and GFP recruitment into PRM-SH3-6His-Ni$^{2+}$ droplets was monitored by measuring client PCs (client fluorescence intensity inside/outside droplets). While free GFP was barely localized into condensates (PC = 1.5), GFP constructs with scaffold components (PRM, SH3, and 6His) were clearly recruited (Fig. 3). GFP recruitment was widely increased as more scaffold components

were fused (PCs: GFP-PRM 3.9, GFP-PRM-SH3 13, and GFP-PRM-SH3-6His 64; Fig. 3b). In particular, PCs were increased most by 6His addition, likely due to stronger Ni$^{2+}$-6His interaction than PRM-SH3 interaction. By simply fusing varying combinations of scaffold components, client can be recruited into the present metal-induced condensates with diverse enrichment power (e.g., PCs range from 1.5–64).

We fused an additional interacting motif on the PRM-SH3-6His scaffold to investigate client recruitment by an orthogonal binding pair. The Glu mutated SpyTag peptide (D7NSpyTag), which non-covalently binds to SpyCatcher ($K_D$ ~1 μM)[49], was fused to PRM-SH3-6His. The recruitment of SpyCatcher-fused GFP to D7NSpyTag-PRM-SH3-6His droplets was highly effective, showing higher PC (26.4) than clients, such as GFP-6His and GFP-PRM-SH3 (Fig. 3). In general, client diffusion inside droplets was slightly faster than scaffold diffusion, and particularly weakly recruited clients showed relatively high diffusivities (Supplementary Fig. 13). The data indicate that the degrees of client recruitment (or enrichment) and diffusivity inside condensates can be controlled by adjusting binding motifs on clients and scaffold proteins.

**Modification of SH3-PRM binding interfaces tunes phase separation tendency.** We next examined how changes on binding interfaces between SH3 and PRM can alter condensate formation processes. SH3 domains typically bind to peptide ligands, which contain a Pro-rich PxxP core consensus-binding motif[38]. Several reported mutational studies on SH3-PRM interactions have shown that substitution of prolines to alanines in PxxP binding motifs reduces binding affinities to SH3 domains[44,50,51]. We mutated two Pro residues of the PRM PxxP binding motif (PTPP) on our PRM-SH3-6His scaffold protein. Even a single Pro-to-Ala mutation (P8A or P11A) dramatically reduced the phase separation tendency, likely due to weakened PRM-SH3 interaction, and therefore, condensate formation requires higher protein and Ni$^{2+}$ concentrations (Figs. 1e and 4a). We measured binding affinities of chemically synthesized PRM mutant peptides to SH3 by isothermal titration calorimetry (ITC). The binding affinity of wild-type PRM was 379 μM, but the affinities of P8APRM and P11APRM were too weak to measure under the present experimental conditions (Supplementary Fig. 14a). Interestingly, the

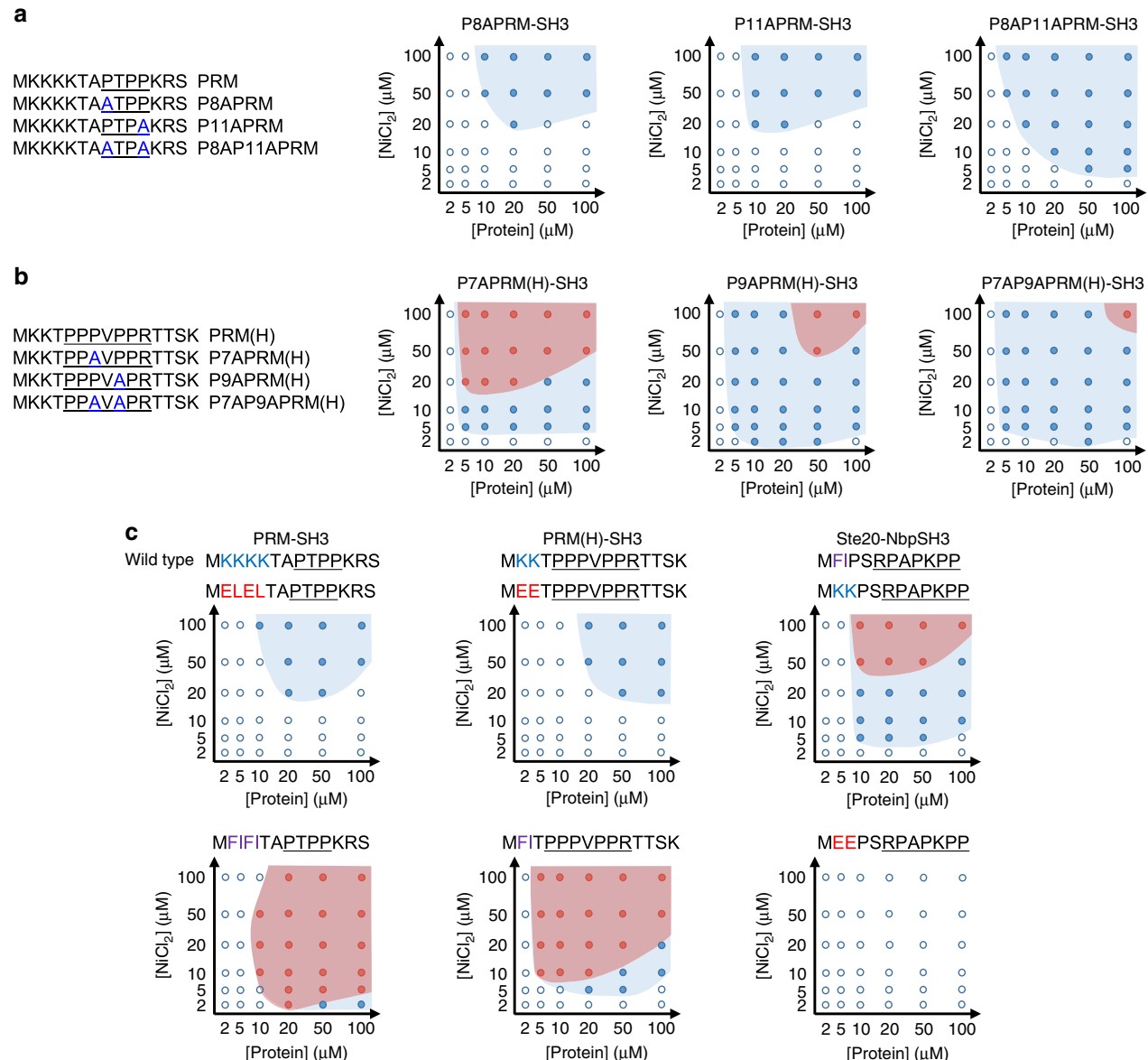

**Fig. 4 Phase separation alternation by modifications of ligand–receptor binding interfaces. a** Phase diagrams of PRM-SH3-6His with Pro-to-Ala mutations. Mutated PRM sequences are shown in the left. **b** Phase diagrams of PRM(H)-SH3-6His with Pro-to-Ala mutations. Mutated PRM(H) sequences are shown in the left. **c** Phase diagrams of ligand-SH3-6His with charge and hydrophobic mutations. Mutated ligand sequences with wild-type sequences are shown in the top of diagrams. Pro-rich core consensus-binding motifs are underlined.

double mutation variant (P8A and P11A) showed a slightly increased phase separation tendency from single Pro mutants, while LLPS still requires higher protein/metal concentrations than wild-type PRM. Pro-to-Ala mutation can increase peptide hydrophobicity to some degree, which might favor phase separation[11]. The data demonstrate how the binding affinity and solubility of scaffold proteins affect phase separation. In addition, inside protein diffusivities of P11A and P8AP11A mutants were slightly higher than wild-type PRM (Supplementary Fig. 15). By applying the present minimal scaffold system, the protein condensation process can be largely altered by even single-residue mutations.

PRM-SH3 binding was fairly weak ($K_D = 379\,\mu$M), and since even weakened PRM Pro-to-Ala mutants can still phase separate by metal clustering, it is possible that strong interactions may not be essential for LLPS. In fact, LLPS processes of IDPs are mostly driven by multiple, but extremely weak residue–residue interactions. We also examined Pro-to-Ala mutants of PRM(H), which

has a higher affinity to SH3. Unlike PRM, threshold protein/metal concentrations were not altered by PRM(H) mutations (Figs. 1e and 4b), which may still have strong enough binding affinities for LLPS even at low concentrations. In fact, the binding affinities of PRM(H) mutations (PRM(H) = 12.8 $\mu$M, P7APRM(H) = 19.4 $\mu$M, P9APRM(H) = 102 $\mu$M, and P7AP9APRM(H) = 131 $\mu$M) were all stronger than PRM (Supplementary Fig. 14). On the other hand, the tendency of PRM(H)-SH3-6His to form more gel-like condensates was noticeably reduced by binding site mutations, particularly for P9APRM(H) and P7AP9APRM(H) variants, with vastly reduced binding affinities. Strong interactions between multivalent scaffold proteins might be one of the contributors to gel-like condensate formation.

We also mutated conserved basic residues nearby the PxxP motif to manipulate receptor–ligand binding interfaces. First, four Lys residues of PRM (PRM-SH3-6His) were mutated to acidic (ELEL) or hydrophobic (FIFI) residues. The acidic ELEL mutant showed a largely reduced LLPS tendency, similar to single Pro-to-Ala

mutated PRM variants (Figs. 1e and 4a, c). On the other hand, hydrophobic FIFI mutation dramatically promoted gel-like condensate formation (Supplementary Fig. 16). We also mutated two Lys residues of PRM(H) (PRM(H)-SH3-6His) again to acidic (EE) and hydrophobic (FI) residues. Similarly, acidic EE mutation reduced the LLPS tendency, and FI mutation showed frequent gel-like condensate formation (Fig. 4c and Supplementary Fig. 16). It has been suggested that conserved Lys residues provide additional bindings by forming salt bridges with acidic residues of SH3 domains[38,52]. Surprisingly, however, both acidic variants showed stronger binding affinities (ELEL-PRM $K_D$ = 121 μM and EE-PRM(H) $K_D$ = 3.32 μM) than their wild-type proteins (Supplementary Fig. 14b), while these Lys-to-Glu mutants show greatly reduced LLPS tendency (Fig. 4c). Charge distribution (as well as hydrophobicity) on scaffold proteins is also likely one of key factors to affect LLPS in addition to binding affinities. These data also explain why different scaffold proteins exhibit vastly different LLPS behaviors (Fig. 1e).

As discussed above, Ste20-NbpSH3-6His did not form any condensates at any protein/metal concentrations. Unlike PRM and PRM(H), Ste20 contains two hydrophobic residues (FI) rather than basic Lys (Fig. 4c). Interestingly, when these FI residues were mutated to basic Lys (KK), clear LLPS was observed at a wide range of protein/metal concentrations, including gel-like condensates at high concentrations (Supplementary Fig. 16), while the EE mutant did not phase separate. The observed LLPS by KK-mutated Ste20-NbpSH3-6His was unlikely due to an increased affinity, since Ste20-NbpSH3 interaction is already strong ($K_D$ = 0.2 μM)[44]. It is still surprising that only two-residue mutations can transform a LLPS incompetent protein into a phase-separable protein, while its working principles remain to be solved.

We also examined structural changes and global unfolding of various scaffold proteins by using a temperature-variable circular dichroism (CD) spectroscopy (Supplementary Fig. 17). Whole scaffold proteins (20 μM) (e.g., PRM-SH3-6His) without metal ions were analyzed with increased temperatures from 20 to 100 °C, whereas ITC was conducted with separated binding peptides (e.g., PRM) and folded globular protein domains (e.g., SH3). Ellipticity profile changes by temperature for free SH3 were well-fitted to a simple two-state transition (folded to unfolded) with a single melting temperature ($T_m$ = 68.8 °C). On the other hand, PRM-SH3-6His data were better fitted to a three-state transition with two $T_m$ values (53.8 and 66.7 °C; Supplementary Fig. 17). Unbinding of PRM-SH3 interactions might be responsible for the first transition, and protein unfolding (likely SH3 unfolding) for the second. CD spectra were obtained for all Pro-to-Ala and acidic mutants of PRM and PRM(H). Structure-dependent ellipticity profiles were mostly unchanged by these mutations. Importantly, $T_m$ changes by mutations were consistent with binding affinity changes (higher $T_m$ for stronger interactions), suggesting that the ITC affinity data could be valid for binding pair-fused scaffold proteins.

**Engineering inter-motif linkers tunes physicochemical properties of protein condensates**. Peptide linkers between multivalent binding receptors and ligands can influence various binding parameters, such as inter-domain distances and flexibility, which can also impact eventual droplet formation and functions. In fact, unfolded linkers between natural multi-domain proteins in the human proteome are highly abundant, and represent a wide range of spacing and flexibility[53]. Previously, the roles of inter-domain linkers between tandemly repeated SH3 (and PRM) units for LLPS processes were theoretically studied. This coarse-grained computer simulation suggested that inter-domain linkers heavily govern phase separation-driven gelation of repeated SH3 and PRM[53].

Here, we intend to provide experimental data on the inter-linker effects during LLPS. The present minimal and single scaffold LLPS system allows easy manipulation of linker peptides. In addition, scaffold clustering by small metal ions will minimally perturb the influence of linkers to droplets. Flexible (GS-rich) linkers and alpha-helix forming rigid (EAAAK-repeated) linkers[54] of various lengths were inserted between the PRM-SH3 domain and the 6His tag (Fig. 5a). Although all linker variants formed protein condensates, higher $Ni^{2+}$ concentrations were needed for LLPS of longer linker scaffolds at the same scaffold concentration (Fig. 5b). For example, a PRM-SH3-Linker-6His variant with a 46 flexible residue linker (FL46) required at least 10 μM $Ni^{2+}$ for LLPS, while only 2 μM $Ni^{2+}$ was needed for LLPS of the original PRM-SH3-6His (FL6). We also introduced long peptide linkers (~20 residues) between PRM and SH3 (PRM-Linker-SH3-6His). The original PRM-SH3-6His contains only three residues (GGS) between PRM and SH3, which can minimize intramolecular PRM-SH3 interactions and offers facile intermolecular interactions for LLPS (Supplementary Fig. 18). Interestingly, PRM-Linker-SH3-6His proteins did not undergo LLPS even with excess $Ni^{2+}$ (50 μM proteins + 200 μM $Ni^{2+}$; Supplementary Fig. 18d). DLS analyses indicated that average PRM-Linker-SH3-6His sizes (3.89 and 4.02 nm) were smaller than that of the original PRM-SH3-6His (4.44 nm). On the other hand, when this long linker was added between PRM-SH3 and 6His (PRM-SH3-FL22-6His), the size (4.91 nm) was larger. These results suggest that the insertion of long linkers between PRM and SH3 promotes intramolecular PRM-SH3 interactions, which increases the portion of collapsed monomer forms (therefore smaller) and inhibits LLPS.

The protein diffusivity of scaffold variants was also noticeably increased as the linker length increased, regardless of the linker types (Fig. 5c). In particular, scaffolds with linkers of >20 residues formed highly dynamic condensates (mobile fraction > 70% with $t_{1/2}$ < 40 s; Supplementary Table 4). Enhanced diffusivity for scaffolds with long linkers might be due to large separation between PRM-SH3 units, which will reduce a multivalent effect and resulting affinity enhancement. Client (PRM-SH3 without 6His) diffusivity inside condensates was also increased with an increase of scaffold linker length (Supplementary Table 5). Relative scaffold enrichment by LLPS of linker variants was examined by measuring PCs of scaffold proteins. PC values were significantly reduced as the linker length increased in a range from 21 (FL6) to 3.1 (RL48) (Supplementary Fig. 19). Client recruitment into droplets was also reduced with increased scaffold linker lengths, while the reduction was only marginal (PCs FL6 = 4.1 and FL46 = 2.0). We also determined phase diagrams with dense/dilute protein concentrations for rigid and flexible long linker variants (RL48 and FL46) with $Ni^{2+}$ or $Zn^{2+}$ (Fig. 5d and Supplementary Fig. 20). Overall, the condensate densities ($C_{dense}$) of FL46 (flexible long linker) were similar to those of original PRM-SH3-6His with only a six residue linker (FL6), indicating tight packing of the FL46 linker. On the other hand, the $C_{dense}$ values of RL46 (rigid long linker) were clearly lower than those of FL6, particularly for maximal $C_{dense}$: RL48 = 31,900–39,000 μM, FL6 = 46,000–51,400 μM, and FL46 = 40,600–57,200 μM. The alpha-helix forming, EAAAK-repeated RL48 (but not flexible FL46) linker might occupy lager volumes than flexible linkers, which lowers maximal condensate densities.

These results indicated that engineering inter-motif linkers in our minimal scaffold modules can tune diverse behaviors of formed condensates, such as LLPS tendency, interior scaffold density, and diffusion dynamics. We envisioned that these condensate behavior changes might also lead to a condensate morphology shift. A long, random, flexible 46 amino acid linker (RFL46) was inserted into PRM(H)-SH3-6His, which easily phase separated to form amorphous gel-like condensates (Fig. 1e).

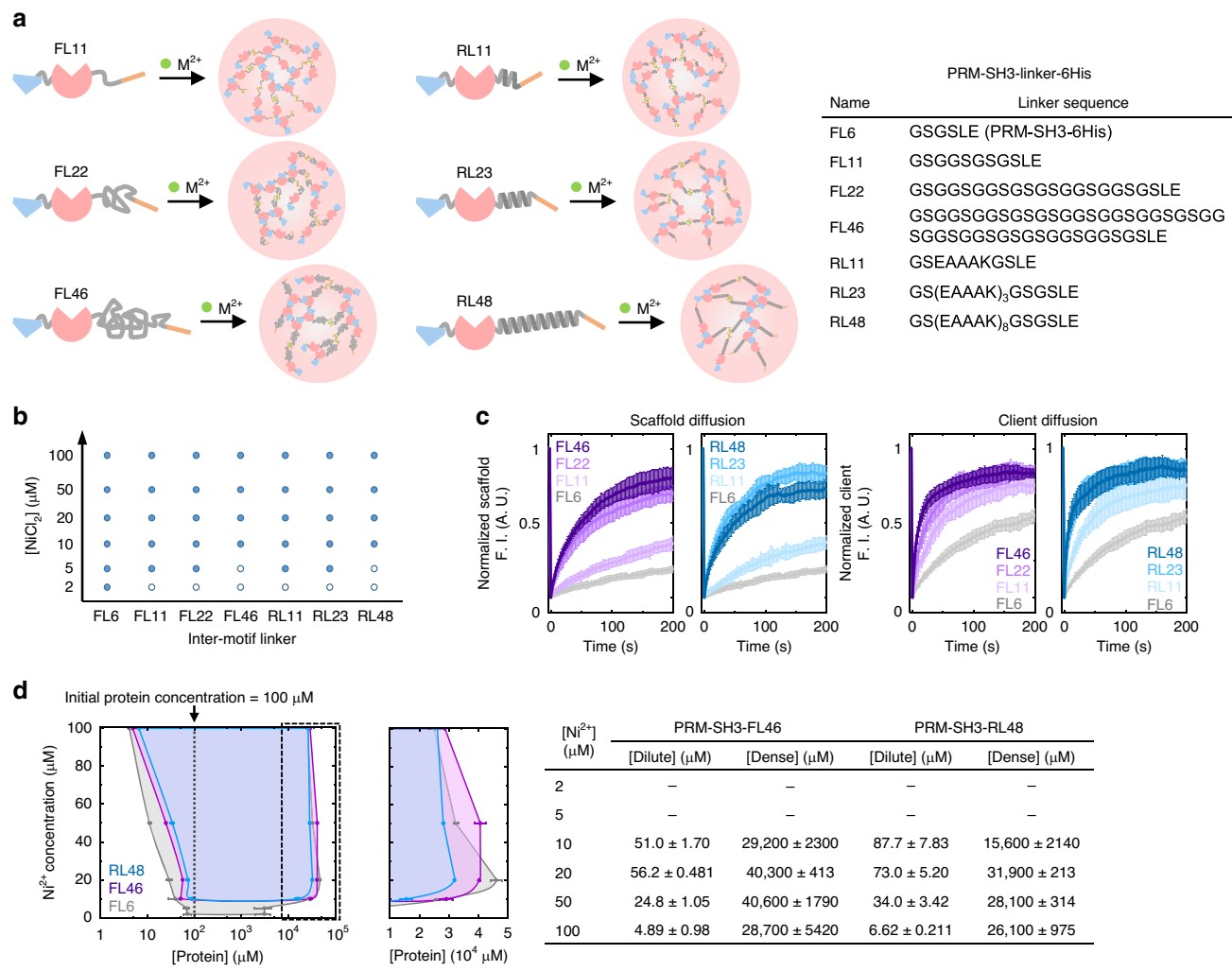

**Fig. 5 Condensate property alternation by engineering inter-binding motif linkers. a** Schematic illustration of metal ion-induced phase separation with linker-modified scaffold proteins. Added linker sequences of PRM-SH3-Linker-6His are shown in the right table. **b** A phase separation diagram of PRM-SH3-Linker-6His as a function of metal ion concentrations. **c** FRAP recovery profiles of PRM-SH3-Linker-6His (left images) and client PRM-SH3 (right images) inside condensates (analysis after 0.5 h upon metal addition). **d** The phase diagrams of PRM-SH3-Linker-6His with dense and dilute phase protein concentrations (analysis after 24 h upon metal addition). The FL6 data are same as Fig. 2e, but included for better comparison with the FL46 and RL48 data. Error bars: 1 s.d. ($n = 2$).

Consistent with the above linker-inserted PRM-SH3, the resulting PRM(H)-SH3-RFL46-6His showed significantly increased threshold concentrations for phase separation, decreased PCs, and enhanced diffusivity inside droplets by linker insertion (Supplementary Fig. 21). Importantly, the formed condensates at all protein/metal concentrations showed spherical liquid droplet shapes rather than amorphous gel-like structures, indicating that linker engineering can also tune the condensate morphology. A recent coarse-grained computer simulation also suggested that spacing properties of linkers between linear multivalent proteins can modulate phase transition behaviors[53].

**Formation of protein liquid condensates can be induced by Zn ions in cells.** We lastly tested whether our metal ion-induced protein condensation can be applied in live cells. PRM-SH3-6His was fused with mCherry and expressed in HeLa cells. For metal ion-induced protein clustering, $Zn^{2+}$ was treated to the cells due to its abundant existence in organisms and low cellular toxicity compared to other metals[55,56]. Upon $Zn^{2+}$ (1 μM) addition, many spherical, micron-sized puncta were clearly observed in mCherry-PRM-SH3-6His expressing cells (Fig. 6a). These cellular

condensates started to appear within only 5 min after zinc addition, and the number of puncta increased over time (Supplementary Fig. 22). When mCherry-PRM-SH3 without 6His was expressed, no puncta were observed (Fig. 6a), indicating that the observed condensates were formed via metal-6His coordination-mediated clustering of phase-separating proteins. Highly mobile protein diffusion of mCherry-PRM-SH3-6His condensates in cells was also confirmed by FRAP (Fig. 6b and Supplementary Table 6). However, when $Zn^{2+}$ was washed out from the cells or even EDTA was treated, the assembled puncta were not disassembled (Supplementary Fig. 23). In addition, protein transfection and $Zn^{2+}$ treatment were somewhat toxic to cells (down to 60% cell viability), although puncta formation was not responsible for this toxicity (Supplementary Fig. 24). Cellular puncta were also observed with PRM(H)-SH3-6His, which formed more gel-like condensates with slow protein diffusion (Fig. 1e, g). Although cellular FRAP analysis provided more fluctuating signal recovery patterns than in vitro FRAP, protein diffusion of mCherry-PRM(H)-SH3-6His droplets was also clearly slower than mCherry-PRM-SH3-6His droplets (Fig. 6c). Nonetheless, it must be noted that metal concentrations are

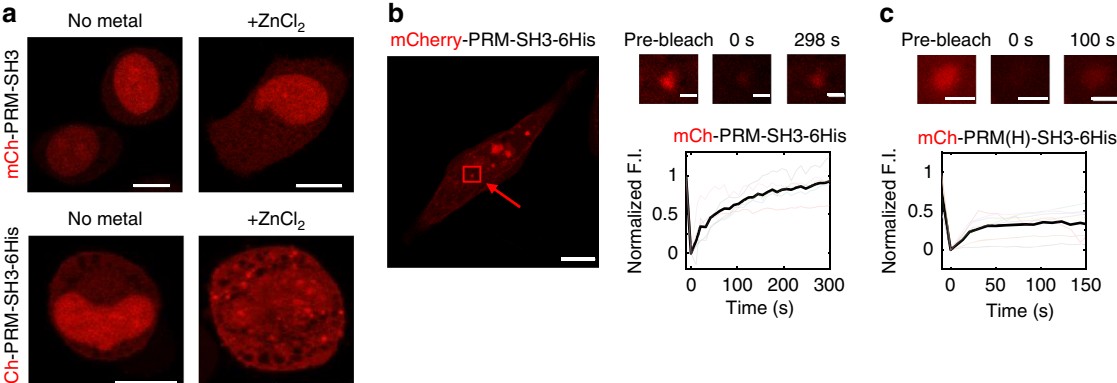

**Fig. 6 Metal ion-induced phase separation in cells. a** Fluorescence images of HeLa cells expressing mCherry-fused PRM-SH3 (top) and PRM-SH3-6His (bottom) scaffold proteins with or without 1 μM $Zn^{2+}$ treatment for 60 min. Scale bars: 10 μm. **b** FRAP recovery profiles and images of cellular mCh-PRM-SH3-6His condensates (scale bars: 1 μm). The whole cell image that contains the bleached condensate is shown on the left (scale bar: 10 μm). **c** FRAP recovery profiles and images of cellular mCh-PRM(H)-SH3-6His condensates. Scale bars: 1 μm.

closed regulated inside cells, and many cellular biomolecules can interact with $Zn^{2+}$.

## Discussion

With simple but versatile 6His–divalent metal ion coordination chemistry, we developed a minimal protein LLPS system with controllable affinities and avidities of scaffold proteins. We demonstrated that various properties of protein liquid condensates can be tuned by applying the present minimal LLPS scaffolds and metal ion clustering. For example, simple changes of added metal ions or metal/protein ratios can alter protein diffusivities of liquid condensates. The degrees of client protein recruitment can also be widely varied by editing binding modules on clients. In addition, linkers between binding modules can be adjusted to control even scaffold protein diffusivities and densities inside condensates. We expect that various molecular and structural codes for protein LLPS can be investigated, as we demonstrated the effects of binding motif spacing on droplet formation. Changes on binding affinities and scaffold solubility heavily influenced protein LLPS. We also observed that only a few residue mutations drastically altered LLPS properties of scaffold proteins. We will continue to test more systematically modified scaffolds to elucidate fundamental rules of multivalent binding module-based protein LLPS. It would also be interesting to examine why only certain binding pairs drive phase separation. In addition, our metal-induced protein clustering can be applied to study LLPS of IDPs for further molecular code investigation. The present method will also be particularly effective to study biochemical processes inside protein condensates. With an ability to control droplet properties, as well as client recruitment, diverse biomolecular reactions under various environments inside condensates can also be examined. Finally, our protein condensates might also offer new biomaterials, which can organize induced condensation of specific sets of biomolecules. This liquid droplet (or diffusive hydrogel) material can be used as in vitro enzyme reaction centers, delivery vesicles, or even as multifunctional protocells.

## Methods

**Protein expression and purification**. All genes encoding indicated proteins were cloned (except the recombinant TEV protease) into the pET-21a expression vector (EMD Biosciences). Cloned plasmids were transformed into *Escherichia coli* BL21 (DE3). The transformed cells were grown at 37 °C until $OD_{600} = 0.6\sim0.8$. Protein expression was induced with 1 mM isopropyl β-D-1-thiogalactopyranoside and incubated at 20 °C for 20 h, except SIM-SUMO-6His and Ste20-NbpSH3-6His variants, which were incubated at 37 °C for 6 h. The induced cells were collected by centrifugation at $6387 \times g$ for 5 min. Harvested cells were resuspended in an equilibration buffer (500 mM NaCl and 50 mM Tris pH 8.0) and lysed by

sonication. Lysed cells were centrifuged at $15,922 \times g$ for 15 min at 4 °C, and 6His-tagged proteins were purified by using Ni-IDA resins (BioProgen). For removal of residual nickel ions in purified proteins, 5 mM EDTA were treated into purified protein eluents, and overnight dialysis at 4 °C was performed twice into PBS at 4 °C. For FIFI/PRM-SH3-6His and FI/PRM(H)-SH3-6His, which prefer to form amorphous gel-like structures, dialysis into PBS was conducted at 25 °C. Final protein concentrations were determined by measuring $A_{280}$, and protein samples were stored at 4 °C, except for FIFI/PRM-SH3-6His and FI/PRM(H)-SH3-6His. To prevent aggregate formation, FIFI/PRM-SH3-6His and FI/PRM(H)-SH3-6His were flash-frozen by liquid $N_2$, then stored at −70 °C. A recombinant TEV protease was produced following the previous protocol[57]. To remove a 6His tag, the purified TEV protease was treated to protein samples at 34 °C for 4 h, and the cleaved 6His tag was removed by using Ni-IDA resins.

**Fluorescent dye labeling of proteins**. Scaffold and client proteins were labeled with cyanine 3 (Cy3) or Cy5 N-hydroxysuccinimide (NHS) ester (Lumiprobe). Protein samples were mixed with cyanine NHS ester in a 1:1 protein-to-dye ratio. The mixed solutions were incubated for 60 min at 25 °C, and dye-conjugated proteins were purified by PD-10 desalting columns (Sephadex™ G-25 M, GE Healthcare).

**Protein phase separation**. Stored proteins were filtered through a 0.2 μm-cellulose syringe filter (DISMIC-13CP, Advantec) before droplet formation. Polyethylene glycol (molecular weight 8000, purity >99%, LPS solution) was added to protein samples (final concentration 5% (w/v)), and the solutions were incubated for 5 min before metal addition. To induce protein phase separation, aqueous $MCl_2$ (M = $Ni^{2+}$, $Zn^{2+}$, $Cu^{2+}$, $Co^{2+}$, $Mn^{2+}$, $Mg^{2+}$, and $Ca^{2+}$) standard solutions were simply added into protein samples to indicated final metal concentrations. Phase separated protein solutions were analyzed on well slides or well plates, which were passivated with 2 mg/mL bovine serum albumin (BSA; Fraction V, Sigma-Aldrich, dissolved in PBS) for 2 h at 25 °C. Phase separation diagrams were obtained by measuring protein solution (100 μL) turbidities (absorbance at 350 nm) on a BSA-passivated 96-well plate with a spectral scanning multimode reader (VarioSkan Flash, Thermo Fisher) after 30 min upon metal ion addition. Proteins were considered to be phase separated when $A_{350}$ differences of metal-added solutions from clear protein solutions without metal ions are >0.03. Phase diagram optical images of formed protein condensates were obtained after 18 h at 25 °C on the well plate. For fluorescent image analyses, 3 μM of Cy5-tagged or GFP-fused proteins were added to scaffold proteins (generally 100 μM).

**Microscopy**. Protein solutions in BSA-passivated flat-bottomed 96-well plates (SPL) were imaged with the Eclipse TS100 inverted microscope (Nikon) using 40×/0.6 numerical aperture (NA) plan-fluorite objective lens. Phase contrast images were obtained and post-processed by the NIS elements 4.0 software (Nikon). For confocal fluorescence microscopy, protein solutions in BSA-passivated eight-well chamber slides (μ-slide eight-well, uncoated, Ibidi) were analyzed by the LSM 800 laser scanning confocal microscope (Carl Zeiss) using 40×/1.40 NA plan-apochromatic oil-immersion and 63×/1.40 NA plan-apochromatic oil-immersion objective lens. GFP-fused constructs were illuminated at 488 nm, and Cy5-tagged proteins were illuminated at 640 nm. Fluorescent and DIC images were taken using the Zen blue software (Carl Zeiss), and collected images were processed and analyzed by the ImageJ software.

**Fluorescence recovery after photobleaching assay**. Circular regions of protein droplets were photobleached, where region of bleaching (ROB) areas are <10% of whole droplet areas. Bleaching was conducted with a 488 or 633 nm laser for GFP-fused or Cy5-tagged protein droplets, respectively. Time-lapse images were taken with a 2 or 5 s period. Droplets with sizes between 10 and 50 $\mu m^2$ were selected. Mean background intensities ($I_{BG}$) were collected from at least ten pixels near the droplet of interest (within 5 $\mu m$ from the outer surface of droplets). We found that fluorescent intensities of droplet centers (targets for bleaching) and other droplet areas can be slightly different (as large as 1.2-fold difference), potentially due to different protein densities over protein droplets on a surface. To compensate this intensity variation and unwanted photobleaching of total droplets, we followed mean fluorescent intensity ratios between regions of bleaching and total droplets. Therefore, relative fluorescent intensity change was monitored by measuring $(I_{ROB} - I_{BG})/(I_{Droplet} - I_{BG})$, where $I_{Droplet}$ is a mean droplet intensity and $I_{ROB}$ is a mean intensity of bleached regions. $(I_{ROB} - I_{BG})/(I_{Droplet} - I_{BG})$ before photobleaching was set to 1, and $(I_{ROB} - I_{BG})/(I_{Droplet} - I_{BG})$ right after photobleaching was set to 0. This normalized fluorescent intensity (nFI) was measured during fluorescence recovery, and recovery curves were fitted to a simple exponential model $nFI(t) = b(1 - e^{-at})$ using the analysis tools Excel® (Microsoft). The recovery half time ($t_{1/2}$) is $ln2/a$, and the mobile fraction is $b$. At least 15 droplets from three independent experiments were selected for the analysis of all diffusions.

**Dynamic light scattering**. Protein solutions and metal chloride standard aqueous solutions were filtered and mixed to indicated final concentrations. After 30 min incubation at 25 °C, 1 mL of the mixed solutions in acrylic disposable cuvettes (Sarstedt) were analyzed (173° backscattered) with a Zetasizer Nano ZS DLS instrument (Malvern Instruments). Each measurement averaged ten runs with a 10 s running time per each run. Data processing was performed with the Zetasizer software.

**Circular dichroism spectroscopy**. CD spectra were measured with a Jasco J-815-150L spectropolarimeter (KAIST Analysis Center for Research Advancement, Daejeon) with Peltier type CD/FL cell holder CDF-426S. Samples were prepared to a final concentration of 20 $\mu M$ in 50 mM sodium phosphate pH 7.4. The ellipticity measurements were performed in a quartz cuvette with 1 mm path length. Thermal unfolding of proteins was monitored by measuring ellipticity at 222 nm wavelength for every 2 °C from 30 to 100 °C with a 5 °C/min temperature increment. All proteins used in this study exhibited complete reversibility upon unfolding. The midpoint transition temperature (melting temperature; $T_m$) of unfolding was calculated by fitting unfolding curves to the van't Hoff equation. A two-state transition model was used for monomeric unfolding $T_m$ value calculation, and A three-state transition model, which contains both unbinding (e.g., interactions between PRM and SH3) and unfolding (e.g., SH3), was used for the unbinding $T_m^1$ and unfolding $T_m^2$ value calculation[58] (see Supplementary Fig. 17 for detailed data analysis).

**Isothermal titration calorimetry**. Isothermal titration calorimetry (ITC) was performed with a MicroCal VP-ITC (Malvern Instruments) instrument. Analyte SH3 domain proteins and titrant chemically synthesized titrant peptides (Peptron) were prepared in PBS pH 7.4, filtered with 0.2 $\mu m$ membrane filter, and degassed with a MicroCal ThermoVac degassing unit. All ITC experiments were performed at 25 °C, and the concentrations in cell and syringe are listed in Supported Information (Supplementary Fig. 14). Total 25 injections were conducted with 10 $\mu L$ per injection and 240 s spacing. The first injection was 2.0 $\mu L$ with 120 s spacing, then the first data point was removed for correct curve fitting. ITC profiles were fitted to a one-site binding model, using the Origin 7 software and the AFFInimiter software.

**Partition coefficient determination**. Confocal images of protein droplets were taken at a specific z-axis focal point, where the total fluorescence intensity of every pixel in an image is maximum. The threshold intensity that defines bulk and droplet phases was determined by averaging two local maxima (most abundant) intensities (a low intensity for bulk phase $I_{Max, Bulk}$ and a high intensity for droplet phase $I_{Max, Droplet}$). The standard deviation of bulk pixel intensities below the threshold ($\sigma_{Bulk}$) was calculated with $I_{Max, Bulk}$ and the highest threshold (($I_{Max, Bulk} + I_{Max, Droplet}$)/2). Mean bulk phase fluorescence intensity was obtained by averaging fluorescence intensities of bulk pixels that are lower than $I_{Max, Bulk} + 1.96 \sigma_{Bulk}$. Similarly, the standard deviation of droplet pixel intensities ($\sigma_{Droplet}$) was calculated with $I_{Max, Droplet}$ and the lowest threshold (($I_{Max, Bulk} + I_{Max, Droplet}$)/2). Mean droplet phase fluorescence intensity was obtained by averaging fluorescence intensities of droplet pixels that are higher than $I_{Max, Droplet} - 1.96 \sigma_{Droplet}$. Droplets with sizes >0.2 $\mu m^2$ were selected. PCs were obtained by dividing inside droplet mean intensities of selected droplets with the outside bulk mean intensities. Between 150 and 800 droplets from at least three independent experiments were selected for PC analysis.

**Quantification of scaffold concentration**. To determine the standard curves for the Cy3-labeled scaffold, Cy3-labeled scaffolds with different concentrations (0–150 $\mu M$) were prepared in a 96-well plate (Thermo Fischer), and fluorescence images were taken using a confocal microscope with 5 and 50% laser powers. Since Cy3-labeled scaffolds were sedimented on surfaces as incubation time increased, fluorescence images were taken at the focal point 200 $\mu m$ above the maximum average fluorescence

intensity, such that the fluorescence intensity was not influenced by bottom surface fluorescence. Two standard curves for each laser power were determined. The first standard curve with 5% laser power was used to determine the dense phase scaffold concentration, and the other curve with 50% laser power was used to determine the dilute phase scaffold concentrations. Unlabeled scaffold and Cy3-labeled scaffolds were mixed with a 3:97 ratio for condensates formation. Condensate solutions were prepared in a BSA-passivated 96-well plate, and then condensate fluorescence images were taken at a focal point with a maximum average fluorescence intensity using a confocal microscope with 5% laser power. Dilute phase samples were taken from the supernatant after sedimentation of condensates samples, then prepared in a 96-well plate. Fluorescence images of dilute phase were taken similarly to the standard curve determination with 50% laser power. Final scaffold concentrations both in condensates and dilute phase were determined by fitting the fluorescence intensity data to the interpolated standard curve and final multiplication with the dilution factor (33.3×).

**Cell experiments**. HeLa cells with fewer than 20 passages were maintained in Dulbecco's Modified Eagles' medium (Gibco) supplemented with 10% fetal bovine serum (HyClone) and 1% penicillin–streptomycin (Gibco) at 37 °C under 5% $CO_2$ in humidified atmosphere. All constructs for cell studies were cloned into the pcDNA3.1(+) vector (Invitrogen). mCh-PRM-SH3-6His or mCh-PRM-SH3-transfected cells were treated with $ZnCl_2$ (1 $\mu M$) and incubated for 0, 5, 30, or 60 min at 37 °C. Cells were washed three times with DPBS and then fixed with 4% paraformaldehyde for confocal imaging. Fluorescence images were obtained with a LSM 800 laser scanning confocal microscope (Carl Zeiss, LSM 800) using 100× oil objective lens. Cell viability was examined by a tetrazolium-based calorimetric assay (MTT (3-(4,5-dimethylthiazol-2-yl)-2,5-diphenyltetrazolium bromide) assay). HeLa cells with fewer than 20 passages were seeded onto 96-well plates at a density of $1 \times 10^4$ cells per well and incubated for 18 h. Transfected cells were treated with various concentration of $ZnCl_2$ and incubated in the medium for 6 h before a MTT analysis.

**Reporting summary**. Further information on research design is available in the Nature Research Reporting Summary linked to this article.

## Data availability
Source data are provided with this paper.

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

## Acknowledgements

This work is supported by Samsung Science and Technology Foundation (SSTF-BA1501-09), BioNano Health Guard Research Center funded by the Ministry of Science and ICT (MSIT) as Global Frontier Project (H-GUARD_2014M3A6B2060507 (1711073453)), and the National Research Foundation of Korea (NRF) grant funded by MSIT (NRF-2019R1A2C2008558).

## Author contributions

Y.J. and H.K. designed the experiments, and wrote the manuscript with help from all authors. H.K. conducted all in vitro experiments. D.S. conducted cellular experiments.

## Competing interests

The authors declare no competing interests.
