## [Peer Review File · Nature Communications]

Reviewers' comments:

Reviewer #1 (Remarks to the Author):

The manuscript entitled “Behavior control of membrane-less protein liquid condensates with metal ion-induced phase separation” by Hong et al. describes multiple designer protein systems featuring well-defined complementary interaction motifs (for example, SH3 and PRM) fused together. Phase separation behavior of poly(SH3) and poly(PRM) proteins as well as client recruitment (individual SH3 and PRM modules) are well documented. Unlike low-complexity IDRs, structural information is available for these modules. Furthermore, several variants of both SH3 and PRM are known that display altered binding affinities.

These factors make these modules attractive as potential scaffolding units to create proteins with tunable phase behavior. In search of a minimal fusion protein, the authors discovered that the His6-SH3-PRM protein phase separation is triggered by low micromolar Ni²⁺ ion.

This observation is consistent with the known coordination chemistry of Ni²⁺ ions and imidazole rings of histidine. In the presence of Ni²⁺ ion, the protein forms multimers that favor LLPS. Therefore, in a way, Ni²⁺ ions increase the valence of the SH3-PRM fusion protein. Cellular studies were also conducted to show that the fusion protein form condensates. This particular observation led a systematic exploration of similar bi-modular systems, such as PRM(H)-SH3-6His, SIM-SUMO-6His, PAK2-βPIXSH3 and Ste20-NbpSH3, and except the last system, the other 4 proteins showed similar condensation behavior.

Interestingly, the authors document some intriguing observations such as gelation for modules with higher interactions affinities, variations in droplet physical properties (protein diffusivity as judged by FRAP; protein partitioning as judged by fluorescence intensity

measurement. The authors also tried to measure the surface tension of some condensates using a confocal based assay. The variations of linker rigidity, as well as length, were conducted and their effect on the LLPS behavior and droplet physical properties were reported. This part of the paper is mostly observational documentation of condensate behavior. Finally, some mutations in the PRM motif were probed to report that mutations can alter droplet formation and their apparent fluid behavior (mostly observational documentation).

At the introduction, the authors say that “precise behavior control of protein droplet models” is desired and that is one of the major motivations for the current study. Although the authors report several interesting pieces of data, in a nutshell, most of their observations are left unexplained. In my opinion, the paper will significantly improve if the following points are taken into account:

a) Page-5: The authors claimed that “Ste20-NbpSH3 did not phase separate at any conditions (Figs. 1e and S3)”. Interestingly, neither of these Figs contain any data on Ste20-NbpSH3 LLPS. Also, these data could not be located anywhere else in the main text and the SI.

b) Page-5: The authors claim “Cu²⁺ is known to have the strongest affinity to His but also to several other amino acids, which might lead to less available Cu²⁺ for 6His-protein clustering” How is this known? There is no reference provided.

c) The authors performed DLS of GFP-tagged PRM-SH3-6His. They claim that the GFP tag prevents LLPS of the protein. However, no data on the phase behavior of this GFP tagged construct has been shown. It is not surprising that GFP will increase the solubility of the protein, but one wishes to know how much is the change compared to the non-GFP tagged protein.

d) The authors probed the condensed phase property with increasing Ni^{+2} . They analyzed droplet size, partition coefficient and diffusivity by FRAP. However, the interpretation lacks much-needed consideration of how increasing Ni^{+2} moves the entire system deeper within a two-phase region. Classically (and analogously), decreasing the Temp of a UCST polymer will produce similar effects due to the increased dimension of the tie lines. With increasing Ni^{+2} , the two phases (dense and dilute) will move further apart in composition. This will change (increase) the relative volume fraction of the dense phase, as well as the density. Together, these changes will produce the effects of increasing Ni^{+2} ions discussed here. Therefore, the results are not surprising but rather expected. Anybody with basic knowledge of LLPs and phase diagrams will predict the observed behavior. One way to improve the analysis here to determine a proper phase diagram and corresponding tie lines (given by the composition of dense and dilute phase at a given salt concentration), which will shed light on why the diffusivity is decreased by increasing Ni^{+2} ions. Is it a linear change with the density of droplets or is there severe chain entanglement (will produce a non-linear effect).

e) Also, the size of the droplets is not the right thing to plot here since the size will be determined by the growth kinetics and the volume fraction of the dense phase. The growth due to coalescence can be slowed down due to increased density at higher Ni^{+2} . It will be better to determine the volume fraction of the dense phase after macroscopically separating the phases or drop this analysis altogether.

f) The time-dependent alterations of condensate fluidity and partition coefficient are very problematic and most likely points towards a strong kinetic component here. The authors did partially acknowledge that. Without suitable care in experimentation and data analysis, this might lead to errors in interpreting the droplet properties, especially when droplet properties between different variants of the same protein are compared, and left the reader

to speculate what if the kinetics of droplet “maturation” is playing a role? I am curious why the authors did not determine if there is a time point that leads to system equilibration. The conclusion in this regard stating “Nonetheless, all these data indicate that physicochemical properties of protein liquid condensates can be tuned simply by adjusting added metal ions, protein/metal ratios, and mixing (aging) times” should be omitted, since there is not enough understanding of the aging process for these systems.

g) The client recruitment by increasing modular interactions (Fig.3) is interesting but not new.

h) The mutation analysis: While the observations are interesting overall, understanding the mechanism of the changes in droplet property due to specific mutation(s) will improve the quality of this paper. Several LLPS papers are out there which suffer from similar low resolution and qualitative observations. Therefore, I strongly encourage the authors to work out a sequence-LLPS model here to explain their results, at least for some of their mutants. These mutant results also clearly suggest that the role of the hexa-His tag is not very simple but should be viewed in light of all possible protein-protein and protein-metal ion interactions.

i) The same criticism holds true for the inter-domain linker studies. Additionally, engineering the inter-domain linker to tune liquid vs Gel has already been studied by the Pappu group (ref 52 in this paper) for similar systems. Therefore, the authors should clearly state the novelty of their findings here.

j) The section describing surface tension measurement is problematic and the conclusion might be only partially right! The surface tension of condensates is not a fixed property but varies based on where you are in the phase diagram. In one sense, the surface tension is proportional to the length of the tie-line (hence the density of the drop). Since phase

separation is altered by changing the linker length, it is likely that the surface tension will also be altered. Even simple Ni-ion variation for the SH3-PRM construct from Fig. 1 or the “aging” will change surface tension. Furthermore, the experiments using contact angle measurement is not properly explained and important refs are missing.

k) Last but not least: condensate is complex fluids. To distinguishing between viscous fluid vs. gels, low-resolution light microscopy (such as Fig S11) might be good for preliminary experiments, but one wishes to see more in-depth results, such as confocal imaging and FRAP. This is true for any distinctions that are made between liquid and gel throughout the paper.

Reviewer #3 (Remarks to the Author):

Hong et al. designed a strategy for metal ion-induced clustering of small protein modules that form biomolecular condensates with liquid-like and gel-like properties. The authors identify that a single polypeptide chain consisting of two interacting domains combined with a 6-His tag establishes the network of interaction for phase separation in the presence of metal ions. They demonstrate that this design principle is valid for several different constructs and compare the ability of the constructs to undergo phase separation. They correlate the ability of the modules to undergo phase separation with several features of resulting interactions, such as the affinity of the metal ions towards the His-tag, as well as the affinity of overall interactions with the material properties of the condensates.

This paper is written very well, and the thoughts and idea are well explained. The experimental approaches are clear, and the analysis and the interpretation of the results are solid. Overall, I feel that this manuscript is well suited for publication. However, I would like to take the opportunity to make some suggestions to the authors.

- From the manuscript it becomes clear that a single polypeptide chain with a His-Tag is more prone to undergo phase separation, compared to previously characterized single chain polypeptide, with repeats of interacting domains in the absence of a His-Tag and metal ions. This suggests that the affinities among the modules are critical (yet weak compared to the his-tag) and that interactions among the His-tags and metal ions can overcome this limitation. Accordingly, the authors mention e.g.: “However, non-spherical and more gel-like condensates were observed at high Ni²⁺ and protein concentrations (Fig. S3), possibly due to stronger PRM(H)-SH3 interaction.”, or “Low protein mobility might be related to their tendency to form more gel-like condensates.” I feel, the authors are in such a position that

they may be able to step away from speculations and may be able to utilize their system to provide a quantitative correlation between the polypeptide affinities (in the presence and absence of ions) and correlate this with phase separation of the system.

- The authors provide DLS data for ion induced clustering using a construct fused to GFP.

The change in rH is rather small, considering that clustering may involve the assembly of multiple monomers to form larger oligomers. Rather than this change being the result of clustering (oligomerization of modules), could this change be rather explained by chain expansion? How do the rH compare to the polypeptide length of a collapsed and expanded monomer? The authors use GFP fused to the construct to suppress phase separation.

Consequently, the GFP inhibits some necessary interactions required for phase separation.

How can the authors be certain that the interactions or chain expansion they observe are relevant and/or on pathway to phase separation? Can the inhibitory effect of GFP be outcompete at high protein concentrations?

- The authors demonstrate that the system ages. Do the condensates become more protein dense with time? The authors should also test if the system is reversible after aging e.g. by addition of EDTA. More importantly, the authors should also test if and to what degree droplet formation is generally reversible by dilution, also after aging.

- I strongly suggest changing the FRAP analysis. The authors decided to normalize the data such that pre-bleach equals 1 and the first datapoint after the bleach equals 0. This removes quantitative information from the analysis, as the deadtime of the experiment is neglected and the number of molecules that become bleached is removed. In ideal cases the recovery amplitude may be indicative for the immobile / mobile fraction when normalized like this.

However, in the experiments provided here, the entire condensate gets bleached in addition to the bleach spot within the condensate. Hence the quantitative relationships

between bleach depth and mobile/immobile fractions are no longer valid. The bleach depth and the total bleach of the condensate should be taken in to consideration.

- The authors could emphasize the role of the his-tag in terms of affinity and avidity.
- The authors utilize the single polypeptide module variants P8A, P11A and double variant P8A/P11A. Since these residues affect proline residues I feel it is important to test for global unfolding and/or conformational changes of the polypeptide. The authors should provide structure data from e.g. circular dichroism and e.g. temperature unfolding in order to correlate the degree of phase separation with the structural stability and folding state of the (unassembled, monomeric) modules.
- Structural stability information should also be provided for other (unassembled, monomeric) modules and variants used in this study.
- The authors provide evidence that the module undergoes phase separation in cells. It is unclear whether the treatment induces cell stress and whether the structures formed by the modules are reversible and/or toxic. E.g. the authors should perform a Zn wash out to test whether the structures disassemble. Moreover, it would be good to know if these structures co-localize with other known condensates, e.g. stress granules.

Titus M. Franzmann

Author's Response to Reviewer #1:

a) Page-5: The authors claimed that “Ste20-NbpSH3 did not phase separate at any conditions (Figs. 1e and S3)”. Interestingly, neither of these Figs contain any data on Ste20-NbpSH3 LLPS. Also, these data could not be located anywhere else in the main text and the SI.

Response: As the reviewer requested, we added a phase diagram of Ste20-NbpSH3-6His (revised Fig. 1e) and optical images of this diagram (revised Fig. S3e).

- In Result and discussion: A Fig. 1e was added.

- In Supplementary Information: A Fig. S3e, which shows optical images of phase separation diagrams of Ste20-NbpSH3-6His, was added.

b) Page-5: The authors claim “Cu²⁺ is known to have the strongest affinity to His but also to several other amino acids, which might lead to less available Cu²⁺ for 6His-protein clustering” How is this known? There is no reference provided.’

Response: We agree that our statement was not justified by proper references, as the reviewer noted. While Cu²⁺ has the strongest affinity to His among tested metal ions (Chem. Rev. 1974, 74, 471), Cu²⁺ also shows the highest binding strength to various proteins (Nature 1975, 258, 598). Still these facts cannot support the statement ‘strong Cu²⁺ binding to other amino acids’. Therefore, we revised the above statement based on the fact that Cu²⁺-His binding is kinetically unstable (fast dissociation). Several studies reported that the dissociation rates of metal ion-His binding follow the order of Cu²⁺ > Zn²⁺ > Co²⁺ > Ni²⁺, while thermodynamic stability follows the order Cu²⁺ > Ni²⁺ > Zn²⁺ ≈ Co²⁺. In fact, when hydrogels were formed through His-divalent metal ion coordination, Cu²⁺-His gel relaxation was faster than Ni²⁺-His gel relaxation (Macromolecules 2013, 46, 1167). Kinetically unstable Cu²⁺-His binding might explain the weaker LLPS tendency of Cu²⁺ than Ni²⁺ (Fig. 2a), despite of strong Cu²⁺-His interactions. In addition, this also better explains the fast diffusion of Cu²⁺-droplets compared to other metal-induced droplets (Fig. 2b).

- In Results and discussion (Page 5, last paragraph – Page 6, first paragraph): This paragraph was revised to better explain LLPS and droplet diffusivity tendencies of the present metal-induced protein condensation (Figs 2a and 2b) with references as discussed above.

c) The authors performed DLS of GFP-tagged PRM-SH3-6His. They claim that the GFP tag prevents LLPS of the protein. However, no data on the phase behavior of this GFP tagged construct has been shown. It is not surprising that GFP will increase the solubility of the protein, but one wishes to know how much is the change compared to the non-GFP tagged protein.

Response: As the reviewer suggested, we examined the Ni²⁺-dependent phase behavior of GFP-tagged PRM-SH3-6His as well as PRM-SH3-6His under the DLS analysis buffer condition (no PEG added). LLPS was not observed by confocal microscopy with up to 500 μM NiCl₂ addition to 100 μM GFP-PRM-SH3-6His, while 50 μM PRM-SH3-6His readily phase separated with 20 μM NiCl₂ (revised Fig. S7).

Additionally, we also conducted DLS analysis for PRM-SH3-6His without GFP to provide information on DLS-based size changes during LLPS. The average size of PRM-SH3-6His/Ni²⁺ complexes increased over 2 μm at the above LLPS condition (50 μM PRM-SH3-6His + 20 μM NiCl₂) (revised Fig. S8c).

- In Results and discussion (Page 6, second paragraph): A sentence “*Under the DLS analysis condition (without a crowding reagent), GFP-PRM-SH3-6His did not undergo LLPS even at 100 μM protein and 500 μM Ni²⁺ (Fig. S7)*” was added.

- In Results and discussion (Page 6, second paragraph): A sentence “*The average size of PRM-SH3-6His without GFP fusion increased over 2 μm by Ni²⁺ addition and presumably subsequent protein condensate formation (Fig. S8c).*” was added.

- In Supplementary Information: A figure (Fig. S7), which shows fluorescence confocal microscopy images of 100 μM GFP-PRM-SH3-6His with different [Ni²⁺], was added.

- In Supplementary Information: A figure (Fig. S8c), which shows the DLS size distribution profiles of PRM-SH3-6His with varying ratios of [Ni²⁺]/[Protein], was added.

d) The authors probed the condensed phase property with increasing Ni²⁺. They analyzed droplet size, partition coefficient and diffusivity by FRAP. However, the interpretation lacks much-needed consideration of how increasing Ni²⁺ moves the entire system deeper within a two-phase region. Classically (and analogously), decreasing the Temp of a UCST polymer will produce similar effects due to the increased dimension of the tie lines. With increasing Ni²⁺, the two phases (dense and dilute) will move further apart in composition. This will change (increase) the relative volume fraction of the dense phase, as well as the density. Together, these changes will produce the effects of increasing Ni²⁺ ions discussed here. Therefore, the results are not surprising but rather expected. Anybody with basic knowledge of LLPs and phase diagrams will predict the observed behavior. One way to improve the analysis here to determine a proper phase diagram and corresponding tie lines (given by the composition of dense and dilute phase at a given salt concentration), which will shed light on why the diffusivity is decreased by increasing Ni²⁺ ions. Is it a linear change with the density of droplets or is there severe chain entanglement (will produce a non-linear effect). condensates they form.

Response: We appreciate this comment and the experimental suggestion. Phase diagrams with dense and dilute phase protein concentrations will be highly valuable to better understand the present LLPS

system. Therefore, in addition to the requested phase diagram of PRM-SH3-6His with increased Ni^{2+} , we also determined phase diagrams with dense/dilute protein concentrations under several other conditions, which will be discussed later. For phase diagrams with corresponding tie lines, we determined the PRM-SH3-6His concentrations of dilute (C_{dilute}) and dense (C_{dense}) phases at varying Ni^{2+} concentrations. As expected from the protein PC values (Fig. 2d) and the reviewer's comment, C_{dilute} decreased as $[\text{Ni}^{2+}]$ increases (revised Fig. 2e), likely because more proteins phase separated into dense phases. On the other hand, C_{dense} rapidly increased as $[\text{Ni}^{2+}]$ increases but peaked at $[\text{Ni}^{2+}]/[\text{protein}] = 0.2$, and then C_{dense} slowly decreased as $[\text{Ni}^{2+}]$ increases. The ratio between clustered PRM-SH3-6His (by Ni^{2+}) and free PRM-SH3-6His will increase with increased $[\text{Ni}^{2+}]$. The data suggest that instead of clustering all PRM-SH3-6His with high $[\text{Ni}^{2+}]$, a certain level of free PRM-SH3-6His is needed to have a maximal C_{dense} (or protein density in droplets) in our multi-component LLPS system (metal ion, clustered PRM-SH3-6His, and free PRM-SH3-6His). A recent study also reported non-linear changes of C_{dense} upon increases of component proteins in multi-component LLPS systems (Nature 2020, 581, 209).

Interestingly, however, the diffusivity of PRM-SH3-6His/ Ni^{2+} condensates was more linearly decreased by increasing $[\text{Ni}^{2+}]$ (revised Fig. 2f). Thereby, it is possible that condensate diffusivities are strongly governed by interaction strengths between scaffold proteins and less by the protein density in condensates. PRM-SH3-6His clustering by metal ions will enhance PRM-SH3 binding strengths by increasing binding valencies. As added $[\text{Ni}^{2+}]$ increases, the population of clustered PRM-SH3-6His will be increased, and clustering stability between PRM-SH3-6His proteins will also increase since more than one metal ions can participate in coordinating hexa-histidines in each protein, which might collectively lead to enhanced scaffold interactions and slowed scaffold mobility.

To further examine this diffusivity and density changes by added metal ions, we also determined the phase diagram with Zn^{2+} (revised Fig. S10), which formed more mobile condensates than Ni^{2+} . With Zn^{2+} , C_{dense} was slightly higher than those of Ni^{2+} -condensates, particularly at high metal concentrations. Since Zn^{2+} interacts more weakly to His than Ni^{2+} , there might be more free PRM-SH3-6His with Zn^{2+} than with Ni^{2+} , yielding more dense Zn^{2+} -condensates. In addition, weak coordination between Zn^{2+} -6His and resulting less PRM-SH3-6His clustering might lead to weak scaffold interactions and fast scaffold mobility of Zn^{2+} -condensates, compared to Ni^{2+} -condensates.

To examine the effects of chain entanglement on diffusivity changes, precise viscoelastic properties (e.g. diffusion coefficient) must be measured, which is still highly challenging for protein condensates. FRAP data offer rather qualitative information. In addition, our LLPS scaffolds consist of transiently interacting small proteins instead of long polymers that are often targets for chain entanglement discussions. More precise and a wide range of analyses on protein droplet structures will be needed to fully understand all physicochemical properties of protein condensates.

Based on the above experimental data and discussions, the section regarding metal concentration dependent droplet property changes was revised as listed below.

- In Result and Discussions (Page 6 last paragraph – Page 7 second paragraph): Two paragraphs were extensively revised based on the above discussions with added figures (Figs 2e and S10) and a reference.
- In Result and Discussions: A figure (Fig. 2e), which shows the phase diagram with $C_{\text{dense/dilute}}$ of PRM-SH3-6His as a function of NiCl_2 and PRM-SH3-6His concentrations, was added.
- In Supplementary Information: A figure (Fig. S10), which includes the phase diagram with $C_{\text{dense/dilute}}$

of PRM-SH3-6His with Zn^{2+} , were added.

e) Also, the size of the droplets is not the right thing to plot here since the size will be determined by the growth kinetics and the volume fraction of the dense phase. The growth due to coalescence can be slowed down due to increased density at higher Ni^{2+} . It will be better to determine the volume fraction of the dense phase after macroscopically separating the phases or drop this analysis altogether.

Response: We agree that droplet sizes (particularly on a surface) were not proper parameters when discussing droplet properties. Instead, measuring protein densities of droplets will provide more valuable information as discussed above. Therefore, we removed the misleading statements and figures regarding the droplet size analysis from the original manuscript, as the reviewer suggested.

- In Introduction (Page 3, second paragraph): In the sentence “*Condensate diffusivity and droplet sizes could be varied by the nature of metal ions (Cu^{2+} , Zn^{2+} , Co^{2+} , Ni^{2+}), incubation time, and protein/metal ion ratios*”, the phrase “*and droplet sizes*” was removed.

- In Results and discussion (Page 6, third paragraph in the original manuscript): The sentences “*Droplet sizes became clearly larger with higher $[Ni^{2+}]$ (Fig. 2d and S7). As the added $[Ni^{2+}]$ increases, a portion of clustered proteins will increase, which will lead to more droplet formation and more droplet fusion to larger droplets.*” and related figures were removed.

- In Results and discussion (Page 6, last paragraph in the original manuscript): A phrase “*Again, the observed droplet sizes on a surface became larger with longer incubation*” and related figures were removed.

f) The time-dependent alterations of condensate fluidity and partition coefficient are very problematic and most likely points towards a strong kinetic component here. The authors did partially acknowledge that. Without suitable care in experimentation and data analysis, this might lead to errors in interpreting the droplet properties, especially when droplets properties between different variants of the same protein are compared, and left the reader to speculate what if the kinetics of droplet “maturation” is playing a role? I am curious why the authors did not determine if there is a time point that leads to system equilibration. The conclusion in this regard stating “*Nonetheless, all these data indicate that physicochemical properties of protein liquid condensates can be tuned simply by adjusting added metal ions, protein/metal ratios, and mixing (aging) times*” should be omitted, since there is not enough understanding of the aging process for these systems.

Response: Again, we agree that the original manuscript lacked proper experimental data and explanation on time-dependent changes of droplet properties, which is critical for better interpreting the droplet behaviors. Therefore, we additionally measured droplet diffusivities by FRAP over 12 h (revised Fig. S11c), instead of original 3 h. Upon Ni^{2+} addition to induce LLPS, the diffusivity was slowly decreased for 1 h but remained constant after 1 h over 12 h, even at various $[Ni^{2+}]$.

Moreover, to examine droplet density changes by time, we also measured dense phase protein concentrations at varying $[Ni^{2+}]$ at 1 h, 3 h, and 24 h after Ni^{2+} addition to induce LLPS. Interestingly, droplet densities were several-fold increased during incubation from 1 h to 24 h (Fig. S11b). Upon

inducing LLPS by initiating metal-6His coordination, the diffusivity of the present condensates reaches equilibrium rather quickly (1 h), while condensate densities are slowly increased.

Interactions between metal ions and proteins, which affect overall scaffold protein binding affinities, might reach an equilibrium within 1 h. On the other hand, tight protein packing in condensates might require more time. The additional data and discussions were added to the revised manuscript. And also we added incubation time-related information for all figures to better illuminate experimental conditions.

- In Results and discussion (Page 7, third paragraph): This paragraph was extensively revised based on the above discussions with added figures (Figs S11b and S11c).

- In Results and discussion (Page 7, third paragraph): A sentence “*Nonetheless, all these data indicate that physicochemical properties of protein liquid condensates can be tuned simply by adjusting added metal ions, protein/metal ratios, and mixing (aging) times*” was removed as the reviewer suggested.

- In Supplementary Information: A figure (Fig. S11b), which shows the phase diagrams of PRM-SH3-6His with dense and dilute phase protein concentrations at different incubation time (t = 1, 3, and 24 h), was added.

- In Supplementary Information: A figure (Fig. S11c), which shows FRAP recovery profiles of PRM-SH3-6His with varying incubation times and Ni²⁺ concentrations, was added.

- In all figure legends: The information on the condensate incubation time after metal addition for LLPS was added.

g) The client recruitment by increasing modular interactions (Fig.3) is interesting but not new.

Response: It is true that the client recruitment by protein condensates is not new as we also cited multiple references. Moreover, enhanced recruitment by increasing modular interactions is also somewhat expected as the reviewer noted. Still, we believe that for newly developed LLPS condensates, providing client recruitment degrees for various constructs will be important information for the proper use of this system. We revised a related statement to better explain the need of these data.

- In Results and discussion (Page 8, second paragraph): A sentence “*Providing quantitative recruitment degrees of various client constructs is important for the proper use of newly developed LLPS systems.*” was added.

h) The mutation analysis: While the observations are interesting overall, understanding the mechanism of the changes in droplet property due to specific mutation(s) will improve the quality of this paper. Several LLPS papers are out there which suffer from similar low resolution and qualitative observations. Therefore, I strongly encourage the authors to work out a sequence-LLPS model here to explain their results, at least for some of their mutants. These mutant results also clearly suggest that the role of the hexa-His tag is not very simple but should be viewed in light of all possible protein-protein and protein-metal ion interactions.

Response: Our initial purpose of this mutation study was to examine the roles of binding interfaces (particularly affinities) between PRM and SH3 on LLPS, while maintaining metal ion-6His clustering. However, as the reviewer noted, our phase diagrams of scaffold protein mutants were largely qualitative observations. To obtain more quantitative information on protein-protein interactions and to better understand the mutation effects, we measured binding affinities (dissociation constants) of many PRM variant-SH3 interaction pairs by isothermal calorimetry (ITC) (revised Fig. S14).

We first measured binding affinities of P-to-A mutants of PRM and PRM(H) against SH3. The binding affinity of wild type PRM-SH3 was 379 μM , but the binding affinities of P8APRM-SH3 and P11APRM-SH3 were too weak (likely over 500 μM) to measure under the present experimental conditions (Fig. S14a). This data suggests that weakened affinities are responsible for the increased critical LLPS concentrations of P-to-A mutants, as shown in Fig. 4a.

K_D values of stronger binding PRM(H) and its P-to-A mutants against SH3 are as follow: PRM(H) 12.8 μM , P7APRM(H) 19.4 μM , P9APRM(H) 102 μM , and P7AP9APRM(H) 131 μM (Fig. S14). PRM(H)-SH3-6His often forms gel-like structures unless [protein] or [metal] is low, which might be explained by the significantly stronger affinity of PRM(H) (12.8 μM) compared to PRM (379 μM). In fact, affinity-weakened P-to-A mutants (particularly P9APRM(H) 102 μM and P7AP9APRM(H) 131 μM) show dramatically reduced tendency for gel-like structure formation (Fig. 4b). These P-to-A mutation results also suggest that even fairly weak protein-protein interactions (like PRM with K_D 379 μM) can induce strong LLPS, at least for the LLPS-favorable PRM/SH3 pair. Strong interactions, however, can cause gel-like condensate formation.

In addition, we also examine charge switching K-to-E mutants. Surprisingly, EELPRM (K_D 121 μM) and EEPRM(H) (K_D 3.32 μM) have stronger binding affinities than their wild type proteins (revised Fig. S14b), while these K-to-E mutants show greatly reduced LLPS tendency (Fig. 4c). Charge distribution on scaffold proteins is also likely one of key factors to affect LLPS in addition to binding affinities. These will also explain why different binding pair scaffold proteins exhibit vastly different LLPS behaviors (Fig. 1e).

To examine global unfolding processes of various scaffold and mutant proteins, we used a temperature-variable circular dichroism (CD) spectroscopy (revised Fig. S17). Whole scaffold proteins (20 μM) (e.g. PRM-SH3-6His) without metal ions were analyzed with increased temperatures, whereas ITC was conducted with separated binding peptides (e.g. PRM) and folded globular protein domains (e.g. SH3). The unfolding curve of free SH3 was well-fitted to a simple two-state transition (folded to unfolded) with single melting temperature (T_m 68.8 $^\circ\text{C}$). On the other hand, the PRM-SH3-6His curve was better fitted to a three-state transition with two T_m values (53.8 $^\circ\text{C}$ and 66.7 $^\circ\text{C}$). Unbinding of PRM-SH3 interactions might be responsible for the first transition, and protein unfolding (likely SH3 unfolding) for the second. CD spectra were obtained for all P-to-A and acidic mutants of PRM and PRM(H). Importantly, T_m changes by mutations were consistent with binding affinity changes (higher T_m for stronger interactions), further supporting the ITC affinity data even for whole scaffold proteins.

The mutation analysis section was revised with these additional K_D/T_m values and the above discussions.

- In Results and discussion (Page 9-11): The section “**Modification of SH3-PRM binding interfaces tunes phase separation tendency.**” was thoroughly revised based on above discussions and added data (Figs S14 and S17).

- In Supplementary Information: A figure (Fig. S14), which contains the ITC thermograms and

calculated binding affinities of PRM variants/SH3 binding, was added.

- In Supplementary Information: A figure (Fig. S17), which contains the CD spectra and calculated T_m values of many scaffold proteins, was added.

i) The same criticism holds true for the inter-domain linker studies. Additionally, engineering the inter-domain linker to tune liquid vs Gel has already been studied by the Pappu group (ref 52 in this paper) for similar systems. Therefore, the authors should clearly state the novelty of their findings here.

Response: Again, to obtain more insights on the roles of inter-domain linkers, we determined phase diagrams with dense/dilute protein concentrations for rigid and flexible long linker variants (RL48 and FL46) with Ni^{2+} or Zn^{2+} (revised Fig. 5d and Fig. S20). Overall, the C_{dense} values of FL46 (flexible long linker) were similar to those of original PRM-SH3-6His with only a 6 residue linker (FL6), indicating tight packing of the FL46 linker. On the other hand, the C_{dense} values of RL46 (rigid long linker) were clearly lower than those of FL6, particularly for maximal C_{dense} : RL48 = 31,900 ~ 39,000 μM , FL6 = 46,000 ~ 51,400 μM , and FL46 = 40,600 ~ 57,200 μM . The alpha-helix forming EAAAK-repeated RL48 linker might occupy larger volumes than flexible linkers, which lowers maximal condensate densities.

Similarly fast FRAP recoveries of RL48 and FL46 variants (Fig. 5c) also support our explanation that binding strengths strongly regulate droplet diffusivities. Upon metal ion-induced clustering, PRM-SH3 units will be further apart by long linkers, which will reduce a multivalent effect and resulting affinity enhancement.

Additionally, we also introduced long peptide linkers (~20 residues) between PRM and SH3 (PRM-Linker-SH3-6His) instead of between PRM-SH3 and 6His. The original PRM-SH3-6His contains only three residues (GGS) between PRM and SH3, which can minimize intra-molecular PRM-SH3 interactions and offers facile inter-molecular interactions for LLPS (revised Fig. S18). Interestingly, PRM-Linker-SH3-6His proteins did not undergo LLPS even with excess Ni^{2+} (50 μM proteins + 200 $\mu M Ni^{2+}$) (Fig. S18d). DLS analyses indicated that average PRM-Linker-SH3-6His sizes (3.89 nm and 4.02 nm) were smaller than that of the original PRM-SH3-6His (4.44 nm). On the other hand, when this long linker was added between PRM-SH3 and 6His (PRM-SH3-FL22-6His), the size (4.91 nm) was larger. These results suggest that insertion of long linkers between PRM and SH3 promotes intra-molecular PRM-SH3 interactions, which increases the portion of *collapsed* monomer forms (therefore smaller) and inhibits LLPS.

Finally, the previous report by the Pappu group was a theoretical study. In our work, we intended to provide experimental data on the inter-linker effects during LLPS. Unlike large multivalent tandem-repeat scaffold proteins, our minimal scaffold LLPS system allowed to facilitate vary these linkers and examine subsequent LLPS. We revised the manuscript to better represent our experimental purpose and novelty.

- In Result and Discussions: A figure (Fig. 5d), which contains phase diagrams of PRM-SH3-FL6, FL46, and RL48 with dilute/dense phase concentrations, was added.

- In Results and discussion (Page 11, second paragraph): Several sentences “*Previously, the roles of inter-domain linkers between tandemly repeated SH3 (and PRM) units for LLPS processes were theoretically studied. This coarse-grained computer simulation suggested that inter-domain linkers*

heavily govern phase separation-driven gelation of repeated SH3 and PRM⁵³. Here, we intend to provide experimental data on the inter-linker effects during LLPS.” were added to better represent our experimental purpose and novelty.

- In Results and discussion (Page 11, second paragraph): Multiple sentences were added to describe inter-domain linkers between PRM and SH3-6His with added data (Fig. S18) as discussed above. - In Results and discussion (Page 12, second paragraph): Multiple sentences were added to describe condensate density changes by inter-domain linkers with added data (Figs 5d and S20) as discussed above.

- In Supplementary Information: A figure (Fig. S18), which contains DLS and LLPS analyses data of PRM-Linker-SH3-6His proteins, was added.

- In Supplementary Information: A figure (Fig. S20), which contains phase diagrams of PRM-SH3-FL6, FL46, and RL48 with dilute/dense phase concentrations (Zn^{2+}), was added.

j) The section describing surface tension measurement is problematic and the conclusion might be only partially right! The surface tension of condensates is not a fixed property but varies based on where you are in the phase diagram. In one sense, the surface tension is proportional to the length of the tie-line (hence the density of the drop). Since phase separation is altered by changing the linker length, it is likely that the surface tension will also be altered. Even simple Ni-ion variation for the SH3-PRM construct from Fig. 1 or the “aging” will change surface tension. Furthermore, the experiments using contact angle measurement is not properly explained and important refs are missing.

Response: We fully agree that surface tension measurement was problematic and can be highly inconclusive. In addition to the inter-motif linkers, various other factors such as added metal ions and incubation time can easily alter surface tensions. Moreover, our C_{dense} measurement further indicate more inconsistency. In fact, contact angle data showed fairly large standard deviations. Therefore, we decided to omit the surface tension data, and replace them with C_{dense} analysis and more inter-domain linker variant studies as discussed above (revised Figs 5e and S18, S20).

- In Results and discussion (Page 11, second paragraph): The sentences and figures related to surface tension measurements were removed.

i) Last but not least: condensate is complex fluids. To distinguishing between viscous fluid vs. gels, low-resolution light microscopy (such as Fig S12) might be good for preliminary experiments, but one wishes to see more in-depth results, such as confocal imaging and FRAP. This is true for any distinctions that are made between liquid and gel throughout the paper.

Response: As the reviewer suggested, we conducted confocal imaging and FRAP assays for non-spherical gel-like condensates. Based on light microscopy, PRM(H)-SH3-6His and PAK2- β PIXSH3-6His formed spherical droplets at low $[Ni^{2+}]$ but formed non-spherical gel-like condensates at high $[Ni^{2+}]$ (Fig. S3). We examined condensates of both scaffold proteins at $[Ni^{2+}] = 10 \mu M$ or $100 \mu M$ (revised Fig. S4). Confocal images of clearly spherical protein condensates were observed at $[Ni^{2+}] =$

10 μM , but non-spherical amorphous images were observed at $[\text{Ni}^{2+}] = 100 \mu\text{M}$. Moreover, protein fluorescence intensities were more heterogeneous for these gel-like condensates. FRAP data also indicate the highly immobile nature of these condensates.

Nonetheless, some of our spherical condensates were also largely immobile (e.g. PAK2- $\beta\text{PIXSH3-6His}$ in Fig. 1g), and discrimination between liquid droplet-like and gel-like condensates is still not clear for protein condensates. In this study, we termed more spherical condensates as liquid droplets and non-spherical amorphous condensates as gel-like structures. Terminology was more defined in the revised manuscript.

- In Result and Discussions (Page 4, last paragraph): A sentence “*Protein fluorescence intensities were rather heterogeneous for these gel-like condensates (Fig. S4a).*” was added.

- In Result and Discussions (Page 5, second paragraph): Two sentences “*In fact, proteins inside these gel-like condensates were nearly immobile (Fig. S4). We termed more spherical condensates as liquid droplets and non-spherical amorphous condensates as gel-like structures.*” were added.

- In Supplementary Information: A figure (Fig. S4), which contains confocal fluorescence images and FRAP recovery profiles of gel-like PRM(H)-SH3-6His/ Ni^{2+} and PAK2- $\beta\text{PIXSH3-6His}/\text{Ni}^{2+}$ condensates, was added.

Author’s Response to Reviewer #3:

• From the manuscript it becomes clear that a single polypeptide chain with a His-Tag is more prone to undergo phase separation, compared to previously characterized single chain polypeptide, with repeats of interacting domains in the absence of a His-Tag and metal ions. This suggests that the affinities among the modules are critical (yet weak compared to the his-tag) and that interactions among the His-tags and metal ions can overcome this limitation. Accordingly, the authors mention e.g.: “However, non-spherical and more gel-like condensates were observed at high Ni^{2+} and protein concentrations (Fig. S3), possibly due to stronger PRM(H)-SH3 interaction.”, or “Low protein mobility might be related to their tendency to form more gel-like condensates.” I feel, the authors are in such a position that they may be able to step away from speculations and may be able to utilize their system to provide a quantitative correlation between the polypeptide affinities (in the presence and absence of ions) and correlate this with phase separation of the system.

Response: To obtain a quantitative correlation between the polypeptide affinities and phase separation of our system as the reviewer suggested, we measured binding affinities (dissociation constants) of many PRM variant-SH3 interaction pairs by isothermal calorimetry (ITC) in the absence of metal ions (revised Fig. S14).

(Also see Response for comment h from Reviewer 1).

We first measured binding affinities of P-to-A mutants of PRM and PRM(H) against SH3. The binding affinity of wild type PRM-SH3 was 379 μM , but the binding affinities of P8APRM-SH3 and P11APRM-SH3 were too weak (likely over 500 μM) to measure under the present experimental conditions (Fig. S14a). This data suggests that weakened affinities are responsible for the increased critical LLPS concentrations of P-to-A mutants, as shown in Fig. 4a.

K_D values of stronger binding PRM(H) and its P-to-A mutants against SH3 are as follow: PRM(H) 12.8 μ M, P7APRM(H) 19.4 μ M, P9APRM(H) 102 μ M, and P7AP9APRM(H) 131 μ M (Fig. S14). PRM(H)-SH3-6His often forms gel-like structures unless [protein] or [metal] is low, which might be explained by the significantly stronger affinity of PRM(H) (12.8 μ M) compared to PRM (379 μ M). In fact, affinity-weakened P-to-A mutants (particularly P9APRM(H) 102 μ M and P7AP9APRM(H) 131 μ M) show dramatically reduced tendency for gel-like structure formation (Fig. 4b). These P-to-A mutation results also suggest that even fairly weak protein-protein interactions (like PRM with K_D 379 μ M) can induce strong LLPS, at least for the LLPS favorable PRM/SH3 pair. Strong interactions, however, can cause gel-like condensate formation.

In addition, we also examine charge switching K-to-E mutants. Surprisingly, ELELPRM (K_D 121 μ M) and EEPRM(H) (K_D 3.32 μ M) have stronger binding affinities than their wild type proteins (Fig. S14b), while these K-to-E mutants show greatly reduced LLPS tendency (Fig. 4c). Charge distribution on scaffold proteins is also likely one of key factors to affect LLPS in addition to binding affinities. These will also explain why different binding pair scaffold proteins exhibit vastly different LLPS behaviors (Fig. 1e).

- In Results and discussion (Page 9-11): The section “**Modification of SH3-PRM binding interfaces tunes phase separation tendency.**” was thoroughly revised based on above discussions and added data (Fig. S14).

- In Supplementary Information: A figure (Fig. S14), which shows the ITC thermograms and calculated binding affinities of PRM variants/SH3 binding, was added.

- The authors provide DLS data for ion induced clustering using a construct fused to GFP. The change in rH is rather small, considering that clustering may involve the assembly of multiple monomers to form larger oligomers. Rather than this change being the result of clustering (oligomerization of modules), could this change be rather explained by chain expansion? How do the rH compare to the polypeptide length of a collapsed and expanded monomer? The authors use GFP fused to the construct to suppress phase separation. Consequently, the GFP inhibits some necessary interactions required for phase separation. How can the authors be certain that the interactions or chain expansion they observe are relevant and/or on pathway to phase separation? Can the inhibitory effect of GFP be outcompete at high protein concentrations?

Response:

Rather than this change being the result of clustering (oligomerization of modules), could this change be rather explained by chain expansion?: As the reviewer pointed, rH changes overall were rather small by metal addition. In particular, the rH change of free GFP-6His by 200 μ M Ni²⁺ addition (~1.5-fold increase, see Fig. S8a) was smaller than the rH change of GFP-PRM-SH3-6His by 200 μ M Ni²⁺ addition (~1.8-fold increase, see Fig. 2c). Although the PRM-SH3 interaction is not strong (K_D 360 μ M), it is still possible that clustered GFP-PRM-SH3-6His proteins can transiently interact with each other for protein chain expansion, leading to slightly higher rH changes. The rH increases of GFP-PRM-SH3-6His by other metals were even smaller (Zn²⁺: ~1.4 fold, Co²⁺: ~1.1 fold, Cu²⁺: ~1.5 fold with 200 μ M metal ions) (Fig. S8b). Weak (Zn²⁺ and Co²⁺) or kinetically unstable (Cu²⁺) binding to

His residues might explain this small rH increases.

- In Results and discussion (Page 6, second paragraph): A phrase was added to a sentence “*Still, multivalent interactions between PRM-SH3 will also contribute to size increases by protein chain expansion, since GFP-6His without PRM-SH3 showed smaller size increases by Ni²⁺ addition (Fig. S8a).*”.

How do the rH compare to the polypeptide length of a collapsed and expanded monomer?: We introduced the minimal GGS linker between PRM and SH3 in our protein construct to minimize the intra-molecular PRM-SH3 interaction, which favors proteins to be expanded forms rather than collapsed (likely due to the intra-molecular interaction of PRM-SH3) monomer. Still, we cannot rule out the possibility that a portion of PRM-SH3-6His exist as collapsed forms, as the reviewer questioned. Therefore, we constructed several new scaffold proteins with long peptide linkers between PRM and SH3, which could alter the tendency of inter- and intra-molecular PRM-SH3 interactions (revised Fig. S18). Interestingly, scaffold proteins with ~20 residue linkers between PRM and SH3 (PRM-FL21-SH3-6His and PRM-RL23-SH3-6His) exhibited smaller rH (3.89 nm and 4.02 nm) than the original PRM-SH3-6His (4.44 nm). On the other hand, when this long linker was added between PRM-SH3 and 6His (PRM-SH3-FL22-6His), rH (4.91 nm) was larger. These results suggest that insertion of long linkers between PRM and SH3 promotes intra-molecular PRM-SH3 interactions, which increases the portion of collapsed monomer forms with reduced rH as schematically described in revised Fig. S18a.

Additionally, we also investigated the phase behavior of PRM-FL21-SH3-6His and PRM-RL22-SH3-6His. Both scaffold proteins did not undergo LLPS even with excess amount of Ni²⁺ addition (50 μM proteins + 200 μM Ni²⁺) (revised Fig. S18d), while PRM-SH3-FL22-6His and PRM-SH3-RL23-6His phase-separated to form condensates (Fig. 5b). These results indicate that long linkers between PRM and SH3 promote collapsed scaffold protein formation, which can inhibit intermolecular protein interaction-based phase separation. In addition, spacing between PRM and SH3 will reduce multivalent effects, which will also reduce LLPS tendency.

- In Results and discussion (Page 11, second paragraph): A paragraph was added to explain collapsed protein construct formation by inserting long linkers between PRM and SH3.

- In Supplementary Information: A figure (Fig. S18), which contains DLS and LLPS analyses data of PRM-Linker-SH3-6His proteins, was added.

How can the authors be certain that the interactions or chain expansion they observe are relevant and/or on pathway to phase separation? Can the inhibitory effect of GFP be outcompete at high protein concentrations?: As the reviewer commented, we are not certain how GFP inhibits phase separation. We suggested that enhanced solubility by GFP fusion might contribute to this inhibition. Under the DLS analysis condition (without crowding reagent), GFP fused PRM-SH3-6His did not undergo LLPS even at 100 μM protein and 500 μM Ni²⁺ (revised Fig. S7). It is also possible that GFP with its relatively large protein size (~25 kDa compared to PRM-SH3 ~10 kDa) might inhibit protein interactions and chain expansion. We added this discussion in the revised manuscript.

- In Results and discussion (Page 6, second paragraph): Two sentences “*Under the DLS analysis condition (without a crowding reagent), GFP-PRM-SH3-6His did not undergo LLPS even at 100 μM*

protein and 500 μM Ni^{2+} (Fig. S7). It is possible that GFP with its relatively large protein size (~25 kDa compared to PRM-SH3 ~10 kDa) might also inhibit protein interactions for LLPS” were added. - In Supplementary Information: A figure (Fig. S7), which shows fluorescence confocal microscopy images of 100 μM GFP-PRM-SH3-6His with different $[\text{Ni}^{2+}]$, was added.

- The authors demonstrate that the system ages. Do the condensates become more protein dense with time? The authors should also test if the system is reversible after aging e.g. by addition of EDTA. More importantly, the authors should also test if and to what degree droplet formation is generally reversible by dilution, also after aging.

Response: As the reviewer questioned, we measured scaffold protein densities at varying time (1 h, 3 h, and 24 h) after metal-induced LLPS (revised Fig. S11b). Interestingly, droplet densities were several-fold increased during incubation from 1 h to 24 h. On the other hand, the condensate diffusivities remained constant during 1 h – 12 h after LLPS (revised Fig. S11c).

Protein condensates were also treated with EDTA after different incubation (aging) times (20 min, 60 min, and 24 h), as the reviewer requested (revised Fig. S12a). Condensates were clearly disassembled (or shrunk) by EDTA, but protein droplets were still visible even after 12 h EDTA incubation, particularly for 24 h-aged (larger) droplets.

These differently aged condensates were also diluted in PBS (revised Fig. S12b). Condensates were mostly resistant against PBS dilution.

- In Results and discussion (Page 7, last paragraph): This paragraph was extensively revised based on the above discussions with added figures (Figs S11b, S11c, S12).

- In Supplementary Information: A figure (Fig. S11b), which contains the phase diagrams of PRM-SH3-6His with dense and dilute phase protein concentrations at different incubation time ($t = 1, 3,$ and 24 hours), was added.

- In Supplementary Information: A figure (Fig. S11c), which contains FRAP recovery profiles of PRM-SH3-6His with varying incubation times and Ni^{2+} concentrations, was added.

- In Supplementary Information: A figure (Fig. S12), which contains confocal images of differently incubated condensates with EDTA treatment or PBS dilution, was added.

- I strongly suggest changing the FRAP analysis. The authors decided to normalize the data such that pre-bleach equals 1 and the first datapoint after the bleach equals 0. This removes quantitative information from the analysis, as the deadtime of the experiment is neglected and the number of molecules that become bleached is removed. In ideal cases the recovery amplitude may be indicative for the immobile / mobile fraction when normalized like this. However, in the experiments provided here, the entire condensate gets bleached in addition to the bleach spot within the condensate. Hence the quantitative relationships between bleach depth and mobile/immobile fractions are no longer valid. The bleach depth and the total bleach of the condensate should be taken in to consideration.

Response: We fully agree that simply setting pre-bleach = 1 can cause various problems, most

importantly unwanted bleaching as the reviewer noted. Additionally, we also found that fluorescent intensities of droplet centers (targets for bleaching) and other droplet areas can be slightly different (as large as 1.2-fold difference), potentially due to different protein densities over protein droplets on a surface. To compensate these variations (unwanted bleaching and heterogeneous droplet intensity), we measured mean fluorescent intensity ratios between regions of bleaching (ROB) and total droplets (I_{ROB}/I_{TOT}). We normalized that I_{ROB}/I_{TOT} equals 1 before bleach and 0 after bleach. We believe that this normalization can minimize biases from whole condensate bleaching and heterogeneous signals of different condensate areas.

However, our original explanation of this normalization was not evident, and thereby we revised the method to provide better description of the present FRAP analysis.

- Methods (Page 15-16, Fluorescence recovery after photobleaching (FRAP) assay): The method was revised to contain the above explanation.

- The authors could emphasize the role of the his-tag in terms of affinity and avidity.

Response: As the reviewer suggested, we revised our manuscript to emphasize the roles of the His-tag (6His) with more information on interactions between His and divalent metal ions. In particular, His interactions with diverse metal ions and resulting effects on avidity changes (and subsequent LLPS) were discussed more with newly added data.

- In Results and discussion, The section “**Metal ions for protein clustering tunes physicochemical properties of condensates.**”: Different interaction properties of four tested metal ions with His and their effects on LLPS were heavily discussed.

- In Conclusions (Page 13): A sentence “*With simple but versatile 6His-divalent metal ion coordination chemistry, we developed a minimal protein LLPS system with controllable affinities and avidities of scaffold proteins.*” was added.

- The authors utilize the single polypeptide module variants P8A, P11A and double variant P8A/P11A. Since these residues affect proline residues I feel it is important to test for global unfolding and/or conformational changes of the polypeptide. The authors should provide structure data from e.g. circular dichroism and e.g. temperature unfolding in order to correlate the degree of phase separation with the structural stability and folding state of the (unassembled, monomeric) modules.
- Structural stability information should also be provided for other (unassembled, monomeric) modules and variants used in this study.

Response: As the reviewer suggested, we examined structural changes and global unfolding of various scaffold and mutant proteins by using a temperature-variable circular dichroism (CD) spectroscopy (revised Fig. S17). Whole scaffold proteins (20 μ M) without metal ions were analyzed with increased temperatures from 20 °C to 100 °C. Ellipticity profile changes by temperature for free SH3 were well-fitted to a simple two-state transition (folded to unfolded) with single melting temperature (T_m 68.8 °C). On the other hand, PRM-SH3-6His data were better fitted to a three-state transition with two T_m

values (T_{m^1} 53.8 °C and T_{m^2} 66.7 °C) (Fig S17). Unbinding of PRM-SH3 interactions might be responsible for the first transition (T_{m^1}), and protein unfolding (SH3 unfolding) for the second (T_{m^2}). CD spectra were obtained for all Pro-to-Ala and acidic mutants of PRM and PRM(H), as suggested by the reviewer. Ellipticity profiles were mostly unchanged by these mutations, indicating structural stability of various binding peptide mutants. Mutations to the binding peptides (PRM and PRM(H)) rather than folded globular domain (SH3) might have only a minimal effects on protein folding/stability.

On the other hand, T_m (particularly T_{m^1}) values were widely varied by mutations. Importantly, these T_m changes by mutations were consistent with our newly added (ITC based) binding affinity changes (revised Fig. S14) (higher T_m for stronger interactions), supporting the statement that T_{m^1} is governed by peptide-SH3 interactions. For example, T_{m^1} values clearly decreased by (binding weakening) P-to-A mutations, while increased by (binding strengthening) K-to-E mutations (Figs S14 and S17).

- In Results and discussion (Page 10, last paragraph): A paragraph was added to explain above discussions with added data (Fig. S17).

- In Supplementary Information: A figure (Fig. S17), which contains CD analysis data of various scaffolds and its mutation variants, was added.

- The authors provide evidence that the module undergoes phase separation in cells. It is unclear whether the treatment induces cell stress and whether the structures formed by the modules are reversible and/or toxic. E.g. the authors should perform a Zn wash out to test whether the structures disassemble. Moreover, it would be good to know if these structures co-localize with other known condensates, e.g. stress granules.

Response: As the reviewer requested, we examined cell viability by MTT assays after scaffold protein expression and Zn^{2+} treatment (revised Fig. S24). Protein transfection slightly reduced cell viability (~80%), and Zn^{2+} treatment was also marginally toxic (down to 60% cell viability). However, no significant loss of cell viability was observed by puncta formation, since cells with mCherry-PRM-SH3 (no puncta) and mCherry-PRM-SH3-6His (puncta) showed similar cell viability.

When Zn^{2+} was washed out from the cells or even EDTA was treated, the assembled puncta were not disassembled (Fig. S23). However, close investigation of our cellular puncta with other known natural condensates has not been possible yet. Metal concentrations are closely regulated inside cells, and many cellular biomolecules can interact with Zn^{2+} . These facts make in-depth characterization (such as measuring Zn^{2+} concentrations) of the present LLPS systems in cells highly challenging. Therefore, in this study (and also during revision), we want to primarily provide comprehensive characterization of the present metal-induced LLPS system in vitro (outside cell), and leave more cellular studies as future tasks.

- In Results and discussion (Page 13, first paragraph): Two sentences “*However, when Zn^{2+} was washed out from the cells or even EDTA was treated, the assembled puncta were not disassembled (Fig. S23). In addition, protein transfection and Zn^{2+} treatment were somewhat toxic to cells (down to 60% cell viability), although puncta formation was not responsible for this toxicity (Fig. S24).*” were added.

- In Results and discussion (Page 13, first paragraph): A sentence “*Nonetheless, it must be noted that metal concentrations are closed regulated inside cells, and many cellular biomolecules can interact with Zn²⁺.*” was added.
- In Supplementary Information: A figure (Fig. S23), which shows the irreversibility of constructed puncta in HeLa cells upon Zn washing or EDTA treatment, was added.
- In Supplementary Information: A figure (Fig. S24), which shows the cell viability of mCherry-PRM-SH3 and mCherry-PRM-SH3-6His overexpressed HeLa cells upon ZnCl₂ addition, was added.

Reviewer Comments:

Reviewer #1 (Remarks to the Author):

The authors have substantially revised the manuscript and added new experimental data based on the previous comments of this reviewer. I would like to congratulate the authors in doing such a great job in revising this manuscript thoroughly. I enjoyed reading the revised paper, especially the newly added phase diagrams. I don't have any more concerns. Thanks!

Reviewer #3 (Remarks to the Author):

The authors addressed my questions and suggestions and discuss the results in the revised manuscript. Accordingly, the new version of the manuscript provides the reader with a more detailed description of the system under investigation.

Titus M. Franzmann